# Revisiting discrete versus continuous models of human behavior: The case of absolute pitch

**Stephen C. Van Hedger**[1,2,3,4]\*, **John Veillette**[1,2]\*, **Shannon L. M. Heald**[1,2], **Howard C. Nusbaum**[1,2]

1 Center for Practical Wisdom, The University of Chicago, Chicago, IL, United States of America,
2 Department of Psychology, The University of Chicago, Chicago, IL, United States of America, 3 Brain and Mind Institute, Western University, London, ON, Canada, 4 Department of Psychology, Huron University College, London, ON, Canada

☉ These authors contributed equally to this work.
\* svanhedg@uwo.ca (SCVH); johnv@uchicago.edu (JV)

**Data Availability Statement:** All data and materials can be accessed through Open Science Framework: https://osf.io/v46aj/.

**Funding:** This research was supported in part from a grant from the John Templeton Foundation

## Abstract

Many human behaviors are discussed in terms of discrete categories. Quantizing behavior in this fashion may provide important traction for understanding the complexities of human experience, but it also may bias understanding of phenomena and associated mechanisms. One example of this is absolute pitch (AP), which is often treated as a discrete trait that is either present or absent (i.e., with easily identifiable near-perfect "genuine" AP possessors and at-chance non-AP possessors) despite emerging evidence that pitch-labeling ability is not all-or-nothing. We used a large-scale online assessment to test the discrete model of AP, specifically by measuring how intermediate performers related to the typically defined "non-AP" and "genuine AP" populations. Consistent with prior research, individuals who performed at-chance (non-AP) reported beginning musical instruction much later than the near-perfect AP participants, and the highest performers were more likely to speak a tonal language than were the lowest performers (though this effect was not as statistically robust as one would expect from prior research). Critically, however, these developmental factors did *not* differentiate the near-perfect AP performers from the intermediate AP performers. Gaussian mixture modeling supported the existence of two performance distributions—the first distribution encompassed both the intermediate and near-perfect AP possessors, whereas the second distribution encompassed only the at-chance participants. Overall, these results provide support for conceptualizing intermediate levels of pitch-labeling ability along the same continuum as genuine AP-level pitch labeling ability—in other words, a continuous distribution of AP skill among all above-chance performers rather than discrete categories of ability. Expanding the inclusion criteria for AP makes it possible to test hypotheses about the mechanisms that underlie this ability and relate this ability to more general cognitive mechanisms involved in other abilities.

## 1. Introduction

In psychological science, human behaviors are sometimes quantized to a small number of discrete categories, often reflected as the simple presence or absence of an ability. Often this is

(58345) to HCN (https://www.templeton.org/). The funders had no role in study design, data collection and analysis, decision to publish, or preparation of the manuscript.

done merely for the sake of empirical tractability, as in the cases of executive function [1], musicality [2] or empathy [3]. In other cases, however, human behavior is discretely categorized by researchers because the boundaries between categories are thought to reflect fundamental differences in the underlying processes and mechanisms that mediate the behavior–an assumption that sharply constrains the types of explanations one can apply to understand such behaviors. A particularly salient example of this is absolute pitch (AP), typically defined as the rare ability to name or produce any musical note without a reference note [4,5].

AP has been described an excellent model system for understanding the interaction of genetic and environmental factors, in part because it is thought to represent a discrete ability that individuals either clearly do or do not possess, which makes the potential genetic and environmental markers theoretically easier to isolate [6]. In fact, the observed dichotomy in note labeling ability in some tests (clusters of near-perfect versus at-chance performers with very few intermediate data points) has led researchers to speculate that AP may be governed by just a couple of genes modulated by early auditory experiences occurring within a critical period [7]. This dichotomized, discrete view of AP suggests that there are two kinds of listeners, clearly differentiated by their explicit abilities to categorize auditory pitch based on chroma. The first kind of listener who, upon hearing an isolated musical note, is essentially guessing its note name, can be thought of as a "non-AP possessor". The second kind of listener who, upon hearing an isolated musical note, can quickly and nearly perfectly categorize the note with its respective note name (e.g., C), can be thought of as a "genuine-AP possessor".

In this vein, describing potential factors that are responsible for the putative dichotomy in note labeling ability has been part and parcel of AP research for the better part of a century [8–10]. One of the most consistently linked experiential factors differentiating AP and non-AP listeners is musical training within a *critical period* of development [11]. Researchers have robustly found that age of musical training onset predicts whether an individual exceeds the experimenter-defined thresholds of performance on pitch labeling tasks that define "genuine" AP ability [8,10–13]. A second experiential factor that has been associated with AP is tone language experience [14]. Several direct assessments of music students enrolled in top music conservatories in the United States and China have found that tone language speakers perform significantly better on AP tests compared to non-tone language speakers and are more likely to exceed typical thresholds for defining "genuine" AP ability [15,16]. While lexical tone distinctions do not depend on absolute pitches (e.g., the "stable" pitch of Tone 1 in Mandarin does not need to be uttered at a particular absolute frequency), the fact that speakers of tone languages show minimal pitch variability in productions–even across days–suggests that absolute pitch cues may be reinforced at least within each speaker of a tone language [14].

Critically, many of the findings concerning experiential contributions to the development of AP depend on a discrete operationalization of AP in their study recruitment design (i.e. two groups of near-ceiling vs. presumably-at-chance performers) and analyses (discrete vs. continuous statistics). Biological phenotypes, however, are often continuous, not discrete, reflecting complex and continuous interactions of genetic and environmental factors that can contribute to the manifestation of a particular trait (e.g., height or weight). When AP is formalized as a discrete ability [7], as much of the literature at least implicitly assumes, direct comparisons of near-perfect AP possessors and at-chance non-AP possessors are appropriate. If, however, AP is a more continuously distributed ability, then such classifications of individuals into "AP" and "non-AP" groups may remove important variance in absolute pitch-labeling ability that may be vital in fully understanding how these experiences relate to the ability.

There are several reasons to suspect that AP is a more continuously distributed ability than previously believed. First, even among self-identified "AP possessors," the distribution of pitch-labeling performance depends on the nature of the test materials, such as familiar versus

unfamiliar timbres [17]. Second, operationalizations of AP that are more graded (e.g., assessing musical note labeling beyond the simple correct/incorrect binary) have shown large amounts of performance variability in both self-identified AP and non-AP populations [18]. Both points highlight potential problems in constructing "AP" and "non-AP" groups for comparison, as the threshold for inclusion in an AP group may be arbitrary and may not capture a significant portion of the population with intermediate pitch naming abilities. While there has been research to suggest that AP may indeed be best described as a continuously distributed ability [5,19], it is presently unclear how particular environmental mechanistic explanations, such as critical-period musical training or tone language experience, relate to *gradations* of absolute pitch-labeling ability. One likely reason why this question has received relatively little attention is because intermediate levels of AP performance (sometimes denigrated as "pseudo" AP) are often treated as an entirely separate phenomenon [20]. Yet, performance thresholds for differentiating "genuine" from "pseudo" AP may not be clear and so these designations may be more reflective of a theoretical framework treating AP as categorically discrete rather than a true separation of ability. Given these considerations, the present large-scale, online study had two primary aims. The first aim was to assess independently whether intermediate AP should indeed be treated as a separate phenomenon from "genuine" AP rather than existing along the same AP continuum. The second aim was to assess how experiential factors (specifically, critical-period musical training and tone language experience) relate to AP performance across different levels of AP proficiency.

The present study uses an online assessment of absolute pitch ability to assess the distribution of absolute pitch labeling ability in the general population. We first apply pitch-labeling performance cut-offs similar to those used in past literature to sort the subjects into "genuine-AP," "pseudo-AP," and "non-AP" groups, and we assess the distribution of experiential factors that have been commonly related to AP (age of music onset and tonal language experience) across these groups. We then apply Gaussian Mixture Modelling to characterize the distribution of pitch labeling performance in a more data-driven manner, identifying two distinct distributions, and we contrast the natural break between these distributions with the normative cut-offs for AP-level. Throughout, we assess whether common findings from past AP research, in which AP performance is predicted by music and tonal language experience, can be explained distribution membership versus continuous position within each distribution, revealing the extent to which the historically dichotomized conception of AP can bias analytic results.

## 2. Method

### 2.1 Participants

We received a total of 195 unique responses (assessed by IP address) to the online AP assessment. These responses spanned a period of approximately 29 months, from May 6, 2016 to October 2, 2018. If multiple responses were associated with a single IP address, we selected the earliest response from each IP to isolate a single respondent. While this culling may appear overly conservative, it should be noted that virtually all the duplicate responses (92.9%) were associated with high levels of note identification accuracy (>80%), suggesting that individuals who performed well were more likely to repeat the measure. Approximately half of the 195 participants (48.6%) completed the study within a one-week period (from June 11, 2017 to June 16, 2017). We attribute this spike in participation to a Wall Street Journal article published on June 11, 2017 [21] that provided a link to the online study.

We culled the data based on three additional considerations prior to analysis. First, we removed participants if they did not complete the brief end-of-study questionnaire, as these

questions were essential for informing our research questions related to age of beginning musical training and tone language background. This consideration removed 25 participants. Second, we removed participants who "timed out" (i.e., did not provide a response) on more than 50% of the trials. This consideration removed an additional 11 participants. Third, we removed participants who reported no musical instruction, as it would not be appropriate to calculate an age of music onset for these individuals. This consideration removed an additional 7 participants. In total, then, 152 participants were included in the analyses. All individuals who completed the online study provided informed consent and the protocol was approved by the Social and Behavioral Sciences Institutional Review Board at the University of Chicago. Participants were not individually compensated for their participation; rather, they had the opportunity to enter (via email) one of several $50 raffles (one drawing per 50 participants).

Participants were not actively recruited by the experimenters to complete the AP assessment (as noted above, most subjects participated following a mention of the test in the Wall Street Journal); thus, we did not employ traditional stopping rules for data collection. Rather, our sample size was based primarily on the length of time the AP assessment had been available online. That being said, it should be noted that the number of analyzable participants (n = 152) is approximately three times larger than a previous study providing evidence that AP is a continuously distributed ability [18]. However, it is smaller than one previous study arguing for AP as a dichotomous ability [7]. Moreover, based on an *a priori* power analysis using G*Power, the present sample size is sufficiently powered ($\beta$ = .8) to detect medium effect sizes even when dividing the sample into three groups; i.e., genuine, intermediate, and non-AP ($f$ = 0.254). The planned analysis that was used in the power analysis was a one-way ANOVA (e.g., assessing differences for age of music onset across groups); the power analysis did not specifically inform the Gaussian Mixture Modeling described in Section 3.4.

## 2.2 Materials

The experimental script was coded in jsPsych (de Leeuw, 2015). We tested absolute pitch-labeling ability with 48 complex tones: 24 "smooth" tones and 24 "triangle" tones. The smooth tones were generated using the "inverted sine" option in Adobe Audition (Adobe Systems: San Jose, CA), which did not result in a true sinusoid but rather a complex tone with 9 harmonics (in addition to the fundamental frequency) and an approximate 11dB reduction for each harmonic. The triangle tones were also generated in Adobe Audition. Both the smooth tones and triangle tones spanned two octaves, from C [3] to B [4]. Additionally, both the smooth tones and triangle tones were 500ms in duration with a 50ms linear onset and offset ramp, and were RMS normalized to -5 dB full spectrum. A representative waveform and spectrum of both the smooth and triangle tone is presented in Fig 1.

## 2.3 Procedure

After providing informed consent, participants were presented with a 10-second sample of pink noise—normalized to the same level as the complex tones—and were instructed to adjust their computer's volume to a comfortable listening level. After this volume adjustment, participants were instructed that they would hear an isolated note on each trial and must identify the name of the note within 5 seconds. After this instruction screen, participants completed the AP test. The 48 notes were presented in a pseudo-random order, as octave always switched between consecutive trials (though the selection of smooth or triangle timbre was random). The choice to continually switch octaves, meant to discourage relative pitch strategies, was influenced by prior research [16], including computerized tests of AP [18]. Participants responded by clicking on one of twelve buttons labeled with a musical note name on the screen

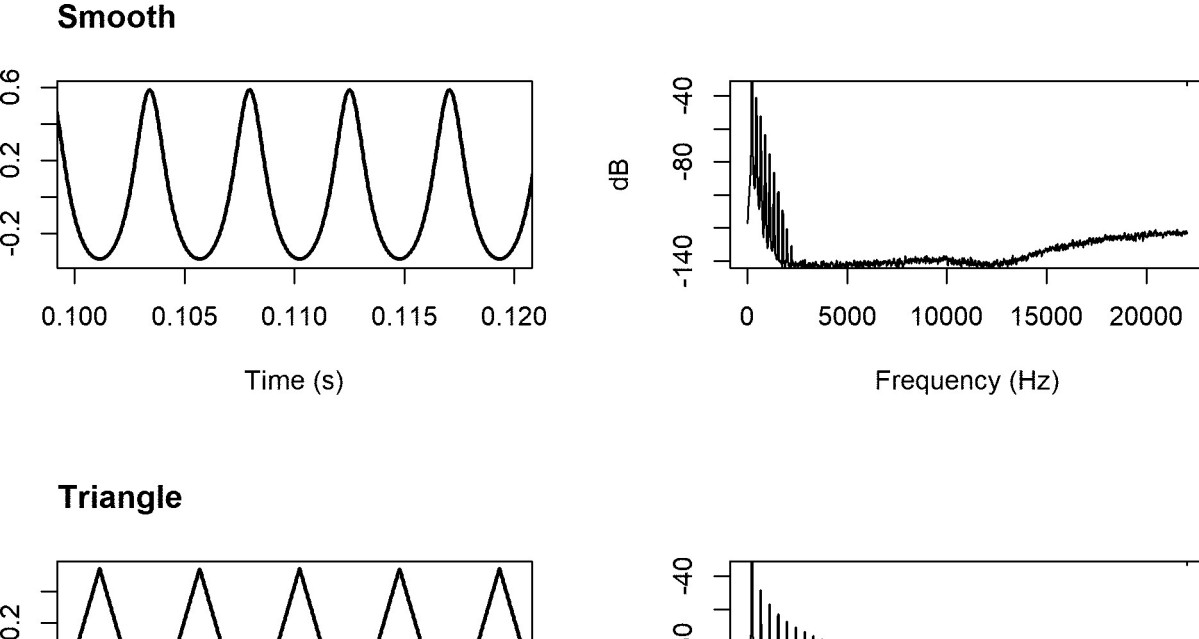

**Fig 1.** Waveform (left) and harmonic spectrum (right) for the smooth tone (top) and the triangle tone (bottom). Plots were generated from actual tones used in the AP assessment (A3; 220Hz).

(e.g., F#). If participants did not respond within 5 second of the note, the trial was marked as a "time-out" and the experiment automatically advanced. As such, once participants advanced to the AP test, there was no opportunity to pause.

After the AP test, participants were presented with a brief questionnaire. Specifically, participants were asked to provide their current age, the age at which they began musical instruction (if applicable), and whether they spoke a tone language. Participants also had the opportunity to provide their email address to be entered into a monetary raffle for completing the study. However, these questions were non-obligatory, and participants could advance beyond this questionnaire page without entering any information. The final screen of the experiment provided participants with feedback (number of correctly identified notes, number of notes missed by one semitone, and percentage correct). The AP assessment took participants approximately five and a half minutes to complete, and most participants completed the entire procedure (from consent to the feedback screen) in under 10 minutes.

## 2.4 Dependent variables

The primary dependent measure we used was a composite score, incorporating both log response time (logRT) and mean absolute deviation (MAD) in semitones from the correct note into a single variable. Both logRT and MAD represent more graded measures of note categorization performance (as they extend beyond the "correct / incorrect" binary), and thus

both are important variables in understanding variability in AP performance. The composite score of logRT and MAD was calculated by adding 10 to an individual's MAD and then multiplying this value by their logRT. For example, if an individuals' MAD was 1.5 semitones from the correct note and their average response time was 3500 milliseconds, their composite score would be 40.76 [(1.5 + 10) * log (3500)]. This composite measure has been previously adopted in AP research [18] and is beneficial in that it imposes a penalty for slower responses.

To assess how our results changed with more restrictive operationalizations of AP, we also calculated a secondary measure. This secondary measure, hereafter referred to as the "conservative measure," adopted the scoring principles outlined in prior influential AP studies [12,22] that have proposed a threshold for exceptional pitch-labeling ability (referred to by the authors as "AP1"). Moreover, this measure has been used previously to support a dichotomy in AP performance [7]. The conservative measure only considered responses made within a 3-second time window (as this was the reported note presentation rate from Baharloo et al. [12]), granting full credit for a correct answer, and granting three-quarters credit for answers that were removed by one semitone from the correct answer. Answers that were removed from the correct answer by more than one semitone were treated equivalently and not granted credit.

Trials in which participants failed to log a response within the 5-second window (i.e., "time-outs") were marked as incorrect and were also assigned a conservative response time of 5 seconds. However, given that no response was logged for time-out trials, MAD could not be calculated.

## 2.5 Data analysis

Our first analysis goal was to characterize our data using the (often arbitrary) performance thresholds that are characteristic of AP research so that we could subsequently compare these groupings to a more data-driven categorization. To this end, we broke subjects into three groups based on their pitch classification accuracies. The break between chance-level, non-AP performers (nAP) and above-chance performers was set at 11 of 48 correctly identified notes (22.9%) because achieving this level of accuracy or higher significantly differed from the chance estimate of 4 of 48 notes (Binomial Test: $p$ = .0003), even when using a false discovery rate alpha correction (FDR q = .0007).

The above-chance performers were further subdivided into pseudo-AP (pAP) performers, whom the AP literature would traditionally not consider performing strongly enough to have AP, and genuine-AP (gAP) performers. The threshold between these two groups was set at 39 of 48 correctly identified notes (81.3%)–a value that was based on prior research. For example, Deutsch et al. [23] used a similar threshold (85%) and operationalized performance both liberally (including semitone errors) and conservatively (only including correct classifications). Furthermore, Miyazaki [24] only tested self-identified AP possessors; however, in this study, the mean accuracy for complex tones (like those used in the present study) was 80.4%. As such, while the placement of any threshold is likely to be arbitrary (assuming a continuous distribution of performance), our selected threshold is grounded in prior research. Whether it is appropriate to separate these two groups at all is tested later in our analyses.

These pitch-labeling thresholds were also balanced with the composite score (incorporating speed of classification and MAD) to determine the final thresholds for the three participant groups. The first group, referred to as "genuine-AP" (gAP), required correctly identifying at least 39 of 48 notes (81.3%) *and a composite measure under 35*. The second group, referred to as "pseudo-AP" (pAP), required correctly identifying at least 11 of 48 notes (22.9%) *and a composite measure under 40*. The accuracy threshold for determining pAP group membership was set at 11 of 48 notes because achieving this level of accuracy or higher significantly differed

from the chance estimate of 4 of 48 notes (Binomial Test: $p$ = .0003), even when using a false discovery rate alpha correction (FDR q = .0007). The third group, referred to as "non-AP" (nAP), encompassed the remaining participants, i.e., correctly identifying fewer than 11 of 48 notes and/or a composite measure of 40 or higher. The gAP and pAP groups were robustly above chance performance, represented by 1/12 (8.33%). The nAP group, in contrast, did not differ from chance. Summary statistics (mean and range) are provided for each group in Table 1.

It is common in the field to compare performance on white vs. black key notes as a measure of AP test sensitivity [18], since decades of research have demonstrated that AP performance is worse on black-key notes compared to white-key notes [16,25,26]. To test whether such a "white-key advantage" was present in the current study, we constructed a 2 (note: black, white) x 3 (group: gAP, pAP, nAP) ANOVA, after which we conducted the appropriate post-hoc tests. This analysis was conducted using both accuracy and the composite measure.

Then, we assessed whether there were differences between these groups with respect to age of musical training onset and tonal language experience. First, we conducted a pairwise $t$-test between the nAP and gAP groups (for age of music onset) and a Fisher's exact test (for tonal language experience) to assess whether we replicate the findings of past AP research when our data are dichotomized in the same way. Then, to assess this relationship for all three groups, we constructed two regression models–one with age of music onset as a continuous response variable and one with tonal language as a binary response variable–using group membership as a regressor. Since some theoretically important inferences could be drawn if the null hypothesis were true, but frequentist analyses do not allow inferences to be validly drawn about evidence for the null hypothesis, we subsequently took a Bayesian approach to comple-ment our frequentist models. To determine the relative evidence in favor of the null versus the alternative hypothesis given the data, we calculated Bayesian equivalents of post-hoc compari-sons among the three groups using JASP 0.9.0.1 (JASP Team, 2018; [27]). In the context of the present analyses, the reported BF (BF01) represents the relative evidence in favor of the *null* hypothesis (i.e., the compared groups do not differ). For example, a BF of 3 would mean that the observed data are 3 times more likely to occur under the null hypothesis, whereas a BF of 1/3 would mean that the data are 3 times more likely to occur under the alternative hypothesis. We conduct these analyses first using the composite measure, and then using the more conser-vative measure described above to ensure our results are robust to the operationalization of AP.

Finally, we characterized the distribution of pitch-labeling performance using data-driven groupings, and we contrasted these groupings with the arbitrary, literature-based groups we had used up until this point. To arrive at these empirical groupings, we used Gaussian Mixture Models (GMMs), implemented with the "mixtools" package in R [28]. By modelling the data as if it is drawn from $k$ normal distributions, we can find the probability of each subject belonging to each group given the normal distributions that best explain the observed data (fit using an Expectation-Maximization algorithm). Importantly, the number of normal

**Table 1.  Summary statistics for the AP groups.**

| Group | *Accuracy* | *MAD* | *RT (s)* |
|---|---|---|---|
| gAP ($n$ = 42) | 93.6% [81.3% - 100%] | 0.05 [0–0.39] | 2.31 [1.60–3.25] |
| pAP ($n$ = 46) | 55.8% [22.9% - 83.3%] | 0.48 [0.06–1.28] | 3.16 [2.11–4.42] |
| nAP ($n$ = 64) | 9.3% [0% - 31.3%] | 2.67 [0.71–3.57] | 3.58 [2.35–4.48] |

Note: Ranges of values are printed in brackets.

distributions used to explain the data is not arbitrarily chosen by the researcher. Rather, this number is determined by a standard model selection criterion of minimizing the Bayesian Information Criterion (BIC). As such, we could assess whether the supposition of three AP groups is empirically supported across operationalizations of AP ability. Since the BIC supported the notion of two distributions, rather than three, we calculated the posterior probability that the predefined pseudo-AP group belonged in the high performing distribution with the genuine-AP performers (as opposed to the other distribution with the at-chance performers); in this way, we empirically test whether pseudo-AP represents a categorically different ability from genuine-AP or if these represent different gradations of the same ability. We again tested two operationalizations of AP performance–the composite measure and the conservative measure.

## 3. Results

### 3.1 AP performance

The distribution of performance, assessed by overall accuracy and the composite measure, is plotted in Fig 2A. "White-key advantage" was present in the current study, indicating that our test was sufficiently sensitive to assess AP performance. For overall accuracy (Fig 3A), we observed a significant main effect of note ($F$ (1, 149) = 33.78, $p < .001$, $n_p^2 = 0.185$), with white-key notes being more accurately identified than black-key notes (white: 49.7%, black: 42.3%). There was also a significant note-by-group interaction ($F$ (2,149) = 10.82, $p < .001$, $n_p^2 = 0.106$), meaning the white-key advantage differed across AP groups. Post-hoc tests showed that this interaction was driven by the pAP group, which had a significantly larger white-key advantage (16.6%) than both the gAP (2.3%) and nAP (4.3%) groups (both $p$s < .001). The attenuated white-key advantage for the gAP group, however, may have been driven by near-ceiling accuracy. Under this explanation, we would expect white-key advantages to manifest in the composite measure, which consists of two non-binary measures and thus may be more sensitive to individual variation.

For the composite measure (Fig 3B), we similarly observed a significant main effect of note ($F$ (1, 149) = 13.63, $p < .001$, $n_p^2 = 0.084$), with white-key notes having lower (better) composite scores than black-key notes (white: 38.88, black: 39.30). The composite measure also showed a significant note-by-group interaction ($F$ (2, 149) = 7.04, $p = .001$, $n_p^2 = 0.086$), with post-hoc tests showing that the gAP (-0.78) and pAP (-0.93) groups had larger white-key advantages compared to the nAP (0.17) group ($p = .014$ and $p = .002$, respectively). These analyses, taken together, suggest that the gAP and pAP groups display worse performance on black key notes compared to white key notes, which is an expected pattern among "AP possessors" [25].

### 3.2 Relationship between early musical training, tone language, and AP performance

The importance of both critical-period musical training and tone language on AP is often supported by directly comparing "AP" and "non-AP" populations, finding that the AP individuals began musical instruction at an earlier age and are more likely speakers of a tone language compared to the non-AP individuals (e.g., see [4,29] for reviews). The present dataset replicates both findings using this dichotomized approach. The gAP group reported a significantly earlier age of music onset compared to the nAP group (5.76 years vs.8.59 years, $t$ (99.1) = 4.74, $p < .001$) and were more likely speakers of a tone language (12 of 42 participants vs. 8 of 64 participants, $p = .046$ Fisher's Exact Test).

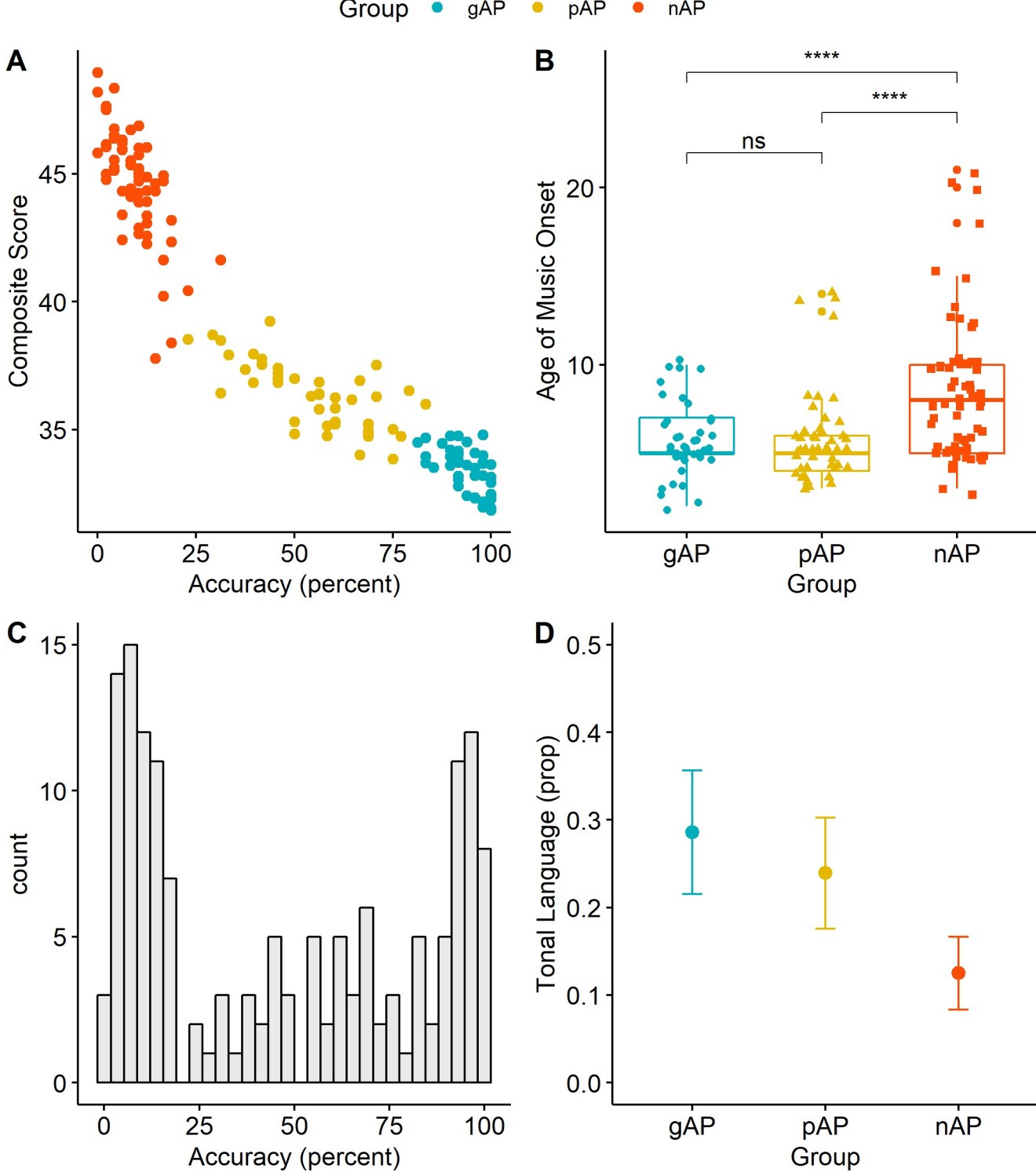

**Fig 2.** Scatterplot of AP performance across all participants, plotting mean accuracy (the percentage of trials in which a subject answered correctly) on the x-axis and the composite score incorporating MAD and logRT on the y-axis (A). The ages at which individuals in each group began musical instruction are shown, with quartiles for each group (genuine-AP, pseudo-AP, and non-AP), in (B). Distribution of pitch-labeling accuracies are shown in (C). Mean proportions of subjects that report speaking a tonal language are in (D), where error bars represent ± 1 standard error of the mean.

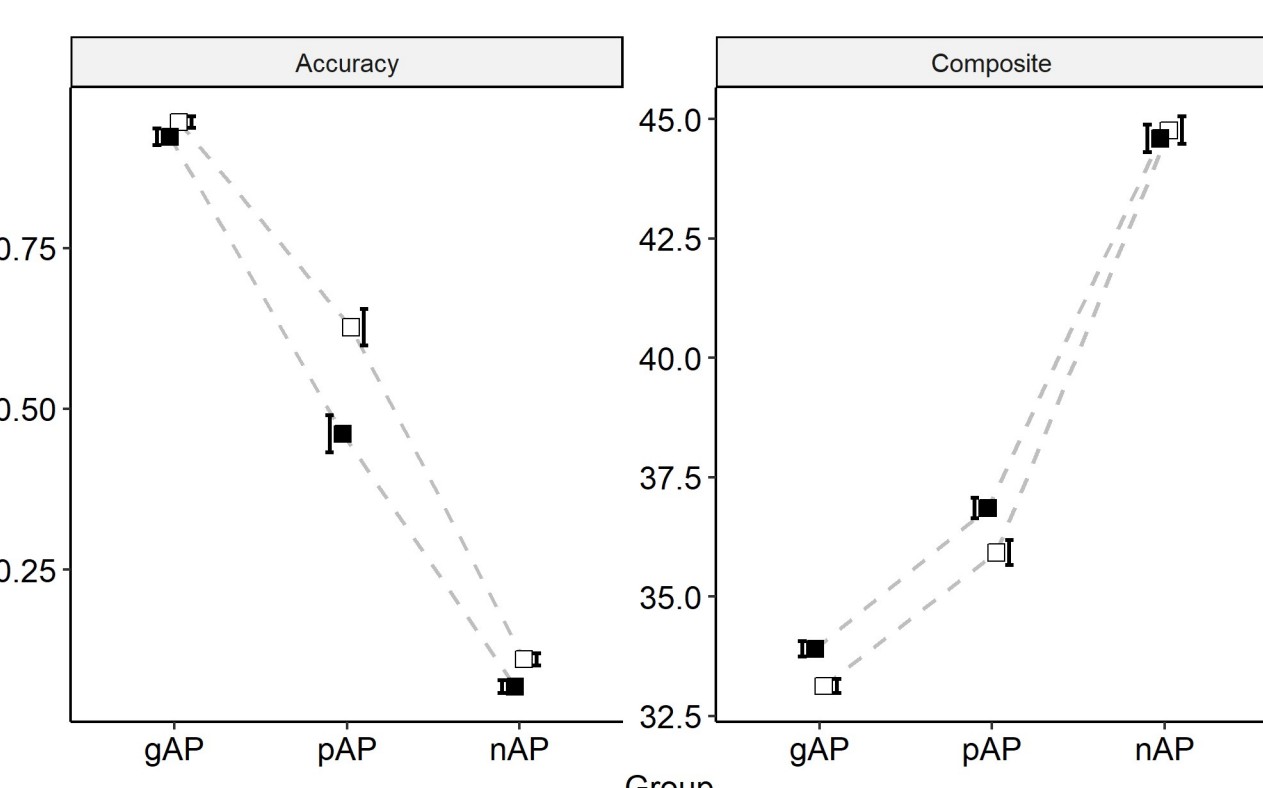

**Fig 3. Interaction plots of from the 2 (note: White, black) x 3 (group: Genuine-AP, pseudo-AP, non-AP) ANOVAs for both mean accuracy and the composite measure.** Error bars represent ± 1 standard error of the mean. Pseudo-AP subjects show the same performance bias known to be characteristic of genuine-AP level performers.

The critical question for the present study, however, is how the pAP group relates to both the gAP and nAP groups in terms of critical-period musical training and tone language experience. The overall regression model for age of beginning musical instruction was significant ($F$ (2, 149) = 13.42, $p < .001$). In terms of group differences, the pAP group reported a significantly earlier age of music onset compared to the nAP group (B = 2.66, SE = 0.62, p < .001) but did not significantly differ from the gAP group (B = -0.17, SE = 0.69, p = .802). Histograms of the age of music onset across groups are plotted in Fig 2B and mean values for each group are plotted in Fig 2C. The overall model for tone language was *not* significant ($F$ (2, 149) = 2.29, $p = .105$). Despite the observation that the pAP group was almost twice as likely to speak a tone language compared to the nAP group (23.9% vs 12.5%), the difference was not significant (B = -0.11, SE = 0.08, $p = .143$). The difference between pAP and gAP participants was also not significant (B = 0.05, SE = 0.09, $p = .587$); however, the pAP and gAP groups were nominally more comparable (pAP: 23.9%, gAP: 28.6%). The mean proportion of tone language speakers across groups is plotted in Fig 2D.

These results suggest that the pAP group and the gAP groups had comparable ages of beginning musical instruction and proportions of tone language proportions, although these findings must be interpreted cautiously as they rest on the acceptance of a null hypothesis. To facilitate appropriate interpretation, since a null result here could be theoretically important but the regression above analyses provides us with little information about the posterior

probability the null hypothesis is correct, we computed Bayes Factors (BF01) to assess evidence in *favor* of the null.

For age of beginning musical instruction, the BF01 for the comparison of the pAP and gAP groups was 4.27, meaning the data were 4.27 times more likely to occur under the null hypothesis. In contrast, the BF01 for the comparison of the pAP and nAP groups was 0.008, meaning that the data were 125 times more likely to occur under the *alternative* hypothesis (i.e., that the pAP group and nAP group differed with respect to their mean age of beginning musical training). For tone language experience, the BF for the comparison of the pAP and gAP groups was 4.03, meaning the data were 4.03 times more likely to occur under the null hypothesis. Unlike the previous analysis, however, the BF01 for the comparison of the pAP and nAP groups was 1.64, meaning that the data were 1.64 times more likely to occur under the null hypothesis. Both of the BF01 values for the pAP/gAP contrasts can be interpreted as providing moderate evidence in favor of the null hypothesis [30], although the tone language results must be interpreted with caution as the BF did not support the alternative hypothesis when comparing the pAP and nAP groups.

### 3.3 Alternative operationalizations of AP performance

One potential concern with the present results is that we have chosen to operationalize AP in a specific manner (i.e., incorporating MAD and log response time into a single composite measure). While this measure has been used in the context of prior AP research [18], it is possible that this composite measure exaggerates the distributed nature of absolute pitch-labeling ability, as both components of the composite measure are non-binary. Given the variability in operationalizing AP across research groups, we thus assessed whether the interpretation of our results would change based on a different operationalization of AP performance.

As a strong test of our observation that AP is continuously distributed among above-chance performers, we chose an alternative operationalization of AP that has been previously used to argue for a dichotomy in AP performance [7]. The present study shares many similarities with the study reported by Athos and colleagues (including computerized testing and the use of non-musical timbres), giving face validity to such an approach. The details of how this conservative measure was calculated is reported in Section 2.4 (*Dependent Variables and Data Analyses*). Timbre is considered separately, meaning the maximum score is 24 (as each timbre consisted of 24 trials). Chance performance using this scoring measure is reflected by a score of 4.75, with the 95% confidence interval around chance performance encompassing the range [2.17, 7.89]. Using an identical equation as reported by the authors of this test to identify the highest ("genuine") AP possessors, participants would need to score above 16.33 to be considered the highest ("AP1") possessors.

We first assessed how our three participant groups–gAP, pAP, and nAP–scored using these performance criteria (Fig 5A). The first thing to note from Fig 4 is that performance on the inverted sine timbre was almost perfectly predictive of performance on the triangle timbre ($r$ (150) = .96, $p < .001$). As such, we averaged inverted sine and triangle scores together for each participant to create a single score (out of 24). Not surprisingly, the gAP group had a high mean score ($M = 19.45$, $SD = 3.24$). However, seven of the 42 participants did not exceed the highest AP threshold of the test, highlighting its conservative nature relative to the composite measure. The nAP group scored essentially at chance ($M = 2.32$, $SD = 1.82$), with no participant exceeding the designated threshold for "AP1" and only one of 64 participants exceeding the upper limit of the 95% confidence interval around chance performance.

Even with this conservative measure, the pAP group scored between the gAP and nAP groups ($M = 10.69$, $SD = 4.66$)–i.e., they did not appear to be compressed toward the non-AP

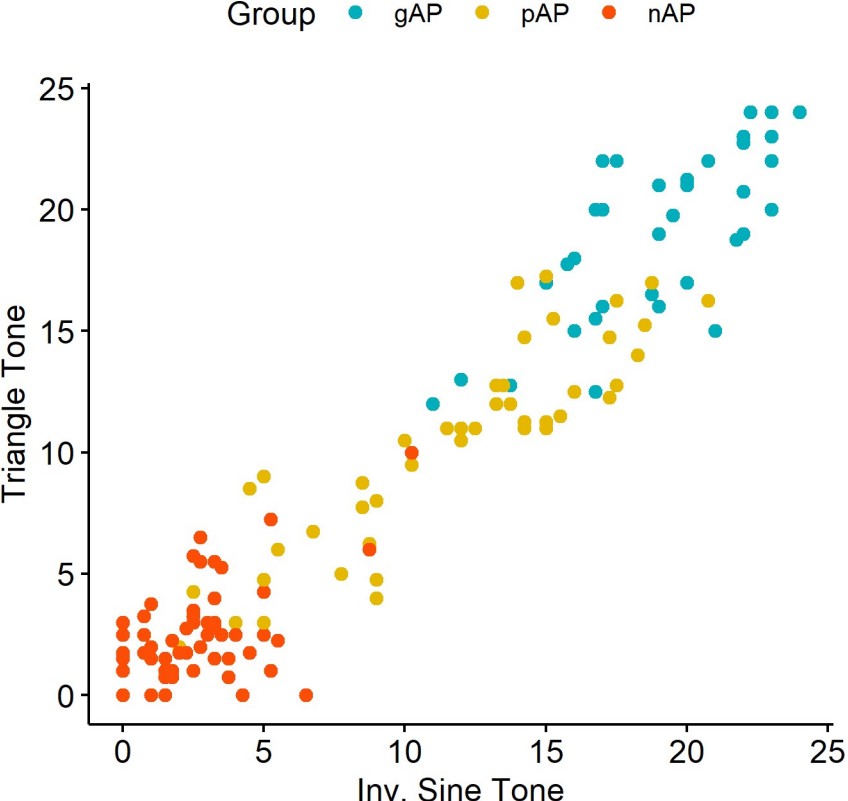

**Fig 4. Scatterplot of performance on the inverted sine (x) and triangle (y) tones, using the conservative scoring measure.** The three groups–genuine AP (gAP), pseudo AP (pAP), and non-AP (nAP) are carried over from the determination made using the composite measure.

end of the distribution, as would be expected in a discrete framework. The pAP level of performance, on average, was almost perfectly between chance performance (5.94 points above) and the "AP1" threshold (5.64 points below). A majority (31 of 46) of the pAP participants exceeded the upper limit of the 95% confidence interval around chance performance, although only a small minority (4 of 42) exceeded the stringent "AP1" threshold. In sum, even using this conservative operationalization of AP, hypothesized to dichotomize performance based on previous work [7], we replicated our findings that AP performance is not discrete, as a sizable proportion of our sample (23.0%) performed at a level greater than expected by chance but below the "AP1" threshold. In fact, the proportion of participants performing at this intermediate level of AP was roughly comparable to the proportion of individuals who were classified as "AP1" (25.7%).

## 3.4 Modelling the distribution of AP performance across measures

While the gAP, pAP, and nAP groupings are useful for facilitating interpretation of our results considering previous research, they nonetheless represent a clear supposition about how AP performance is distributed. To mitigate the possibility of such arbitrary grouping influencing the interpretation of our results, we conducted further analyses using empirical, rather than *a priori*, groupings of the data. Since Bayesian Information Criteria supported two groups for both AP operationalizations tested, we were able to assess whether intermediate performers (pseudo-AP) were more naturally grouped with genuine-AP performers, suggesting that pseudo-AP and genuine-AP are just gradations of the same ability, or with the low performers.

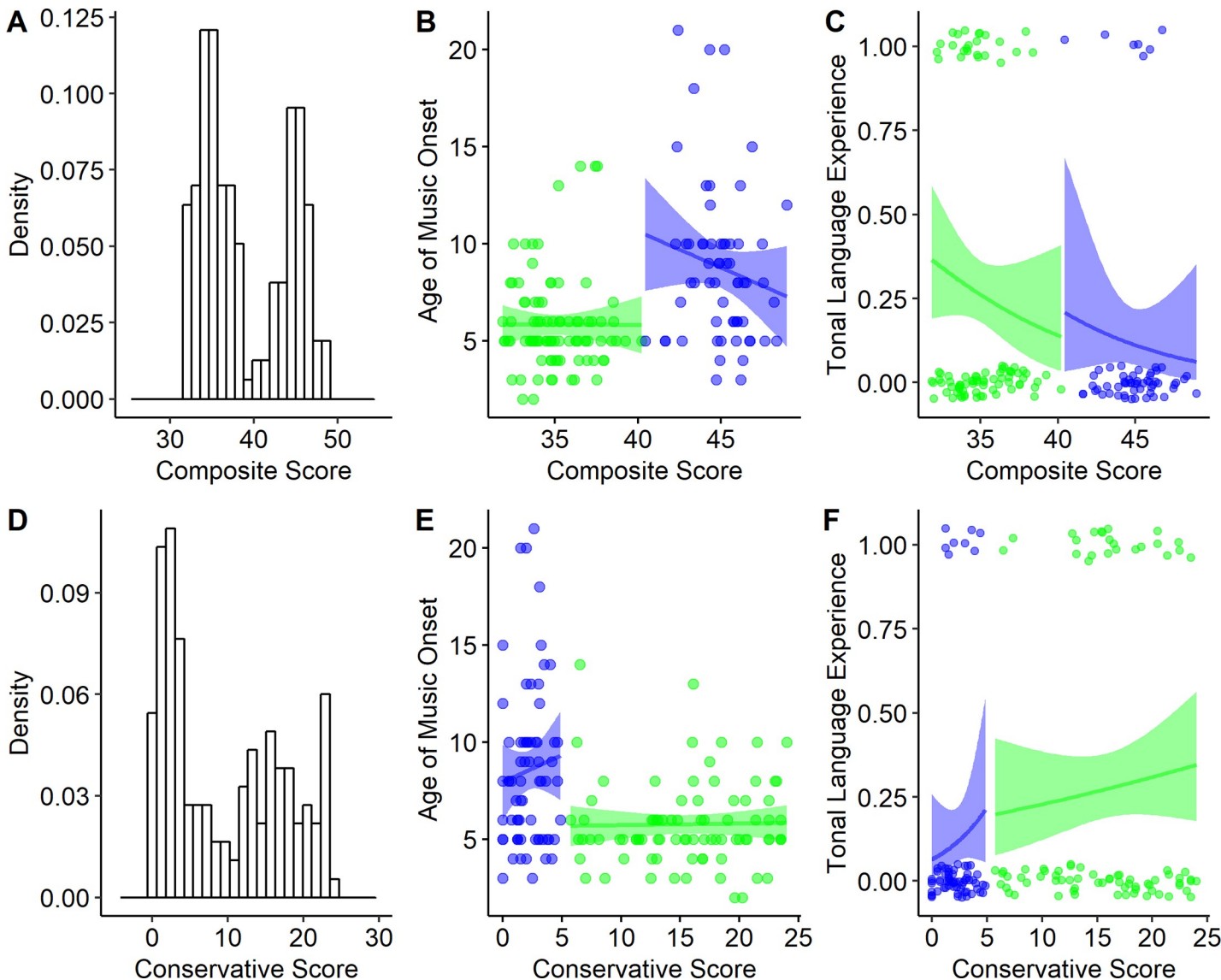

**Fig 5. Results from the distributional modelling of AP performance using Gaussian Mixture Models (GMMs). 5A:** GMM results from the composite score, depicting the best-fit distributions of the high-performing (green) and low-performing (blue) groups, overlaid on a histogram of the AP composite score. **5B:** Age of musical training onset plotted against composite score; points are colored according to the distribution they are more likely to belong to (posterior probability > 0.5), and lines of best fit are shown for each distribution. The line is virtually flat for the high performing distribution, suggesting that age of beginning musical training does little to differentiate gradations of pitch-labeling ability, consistent with the results shown in Fig 2. **5C:** Tonal language experience vs. composite score; logistic best fit, representing the probability of speaking a tonal language given the composite score, is shown for each distribution. **5D-F:** The same plots as 5A-C but using the more conservative operationalization of AP instead of composite score.

**3.4.1 Composite score.** For the composite score, the best-fitting model only included two groups. The increase in BIC from a two- to three-group model was 6.19. Applying a two-group GMM to the composite scores, we find that the distribution of the scores are best described as a mixture of the two normal distributions $N(35.06, 3.66)$ and $N(44.99, 3.04)$, shown in Fig 5A. The normal distribution with the lower mean composite score (representing superior performance) is taken to represent the AP group, and the distribution with the higher mean score is taken to represent the non-AP group. An advantage of GMMs is that they allow for overlapping clusters, but our model nonetheless makes group assignments with a high degree of

certainty, suggesting that the AP and non-AP distributions do not overlap substantially. Interestingly, these two distributions are well aligned with the mean composite scores of the AP (35.81) and non-AP (46.92) groups described in Bermudez and Zatorre [18].

Given that a two-group model, on the surface, may be thought to support a dichotomy in AP performance, the critical question is how the intermediate performers (i.e., the pAP group) is represented within these two distributions. To this end, a convenient summary statistic to use is the average probability of someone who is typically considered to have pseudo-AP performance coming from the AP group in our model. This average posterior probability of being part of the AP group rounds to 1.00 for the pAP participants (0.999), supporting the notion that pseudo-AP possessors are drawn from the same distribution as genuine AP possessors rather than the non-AP distribution. While this number should, by no means, be taken as the probability of the continuously distributed (among above-chance performers) hypotheses being true–indeed, there are many alternate hypotheses and distributional forms not considered here–it resoundingly favors the hypothesis that in two-group characterizations of AP, intermediate performance is better explained as part of the more accurate distribution of ability (including "genuine AP" performers), as opposed to the less accurate distribution of ability (encompassing at-chance, "non AP" performers).

**3.4.2 Conservative measure.** For the conservative measure, the best-fitting model was also a two-group model, which is perhaps not surprising as this measure has been previously used to support a dichotomy in AP. The three-group model provided a worse fit (increase in BIC of 8.22). Applying a two-group GMM to the conservative measure, we find that the distribution of the scores are best described as a mixture of normal distributions $N$ (2.18, 1.36) and $N$ (15.28, 5.50), shown in Fig 5B. These two distributions nominally can be thought of as a worse-performing, "non-AP" distribution, and a higher-performing, "AP" distribution. Similar to the composite measure analysis, we assessed the posterior probability of the pAP group being represented in the AP distribution. While the results were less decisive than the composite measure, the average posterior probability of belonging to the AP distribution for pAP participants was 0.8467, similarly supporting the notion that pseudo-AP possessors are more likely to belong to the same distribution as genuine AP possessors compared to non-AP possessors.

## 4. Discussion

Accurate descriptions of behavior are fundamental to developing explanations of the behavior. Thus, the way a behavior is distributed in the population is critical for informing any theory of underlying mechanisms and subsequent advances in characterizing the phenotype. When AP is conceptualized as simply present or absent, explaining AP requires theoretic mechanisms that are themselves rarely invoked and must combine in such a way as to materialize the ability without variability. Moreover, this discrete framework shapes the basic scientific questions in particular ways, limiting the range or characterization of participants who are included in research.

The present results directly challenge the hypothesis that AP is a categorically discrete ability, characterized by virtually no performance variability between clusters of traditionally defined "AP" and "non-AP" participants [7]. To highlight this point, 52 (30.6%) of all respondents performed with sufficient speed and accuracy to be distinguished from chance performance yet not considered to represent "genuine" AP ability based on commonly adopted thresholds (e.g., minimum of 85% accuracy). Simply discarding these participants and only comparing the extremes of the AP spectrum provided support for role of critical-period musical training and tone language experience in pitch-labeling, replicating prior research in terms

of potential mechanistic explanations of AP. Importantly, however, the present results clearly demonstrate that intermediate levels of AP look more like "genuine" AP in terms of the experiential signatures of critical-period musical training, tone language experience, and even internote performance differences such as white-key advantages. As such, these factors appear to relate to AP performance, but they derive their predictive value in differentiating "at-chance" from "above-chance" individuals, not the "genuine" AP possessors from all other individuals.

The stark dichotomy proposed in the case of AP—which is at least tacitly endorsed in AP research through the paradigmatic contrast of "AP" and "non-AP" groups as representative categories—has shaped the kinds of questions and results that have been reported. For example, studies examining the extent to which AP can be explicitly trained have generally supported the idea that AP performance can be improved in both children [31] and adults [32–35] to varying degrees. Yet, these demonstrable performance improvements are not seriously considered in the context of "genuine" AP ability–particularly among adults–given the discrete framing of AP in combination with critical period theories of AP development. In essence, these factors treat AP as a fixed or crystallized talent, and thus any meaningful improvements to pitch-labeling ability in adulthood must be explained through alternate mechanisms that mimic–but are distinct from–those underlying AP.

Moreover, this approach that has shaped much of the field is puzzling given the suggestions that there are likely multiple factors that contribute to the development of AP [36,37], which would suggest a more continuous distribution of ability. Furthermore, the finding that "AP possessors" tend to use multiple strategies in categorizing musical notes [38] supports the notion that, even among a conventionally-defined AP group, there may be significant variability in strategies and even overall ability. Understanding that AP varies within and between individual listeners is much more consistent with a cognitive process or skill [39] than to a kind of genetic endowment or a holistically acquired trait or talent.

While recent research supports the treatment of AP as a *continuously* distributed ability [17,18], intermediate AP abilities have received relatively little empirical attention in the broader context of AP research. Why might this be the case? One practical reason is that individuals with intermediate AP may not strongly self-identify as "AP possessors," biasing recruitment efforts. One theoretical reason is that the discrete view of AP implicitly informs the design, analysis, and selection of participants for many AP studies. For example, intermediate levels of AP performance may be underrepresented due to how AP performance is tested and scored (e.g., treating all incorrect answers as equivalent, not modeling response time). Beyond biased recruitment and testing, even when intermediate AP is experimentally assessed, it is often assumed to represent a fundamentally different phenomenon than "genuine" AP [20,40] -ostensibly served by different mechanisms—rather than existing along the same AP continuum. By treating all individuals who do not reach this threshold as equivalent, it becomes difficult to understand the true distributional nature of AP, particularly if there is a substantial number of individuals who demonstrate intermediate AP performance across a variety of AP operationalizations (as in the present study).

Conceptualizing AP as a continuously distributed ability among above-chance performers may provide insights into understanding the nature of pitch memory in humans more broadly. In particular, a growing body of research has demonstrated that "non-AP" possessors display good absolute pitch memory for familiar sounds in their environment, such as popular music recordings [41–47] and even non-musical, pitched sounds, such as the North American landline dial tone [48]. This kind of pitch memory, sometimes referred to as *implicit AP*, *latent AP*, or simply *pitch memory* [44,49,50], appears to be normally distributed in the population and has also been thought to represent a foundation for "genuine" AP (i.e., exceptionally accurate pitch memory may scaffold explicit pitch labeling). Supporting this idea, research has

demonstrated that AP possessors have more accurate "implicit" pitch memories compared to musically-matched controls [51], even when judging a non-musical stimulus in which their explicit note labels would not be beneficial [52]. Moreover, this pitch memory may reflect the effectiveness or precision of auditory working memory, given that auditory working memory appears to predict both the accuracy of "implicit" pitch memories for familiar recordings [47] (Van Hedger et al., 2017) as well as the explicit ability to learn AP categories [34].

Implicit pitch memory also appears to generalize beyond the specific recordings heard in one's listening environment, which suggests a more abstracted representation of pitch chroma and provides a compelling connection to explicit AP. For example, non-AP listeners appear to respond differentially to isolated notes based on how frequently the notes are heard in the listening environment [53,54]. Relatedly, non-AP listeners can differentiate conventionally "in-tune" from "out-of-tune" notes with above-chance accuracy, though musicians outperform non-musicians and the effects are not observed for unfamiliar timbres [55]. These findings point to the intriguing possibility that the long-term representation of pitch chroma in memory generalizes beyond specific auditory experiences. The primary difference between pitch chroma representations between AP and non-AP listeners, then, may lie in the resolution of the representations, with "genuine" AP possessors displaying precise representations and non-AP possessors displaying broader representations (e.g., see [29]). It should be emphasized, however, that this kind of framework (differences in the resolution of pitch chroma) appears to be better represented as a continuum of ability.

Thus, the treatment of explicit AP labeling as an ability that is continuously distributed among above-chance performers opens an interesting line of potential research that may help to integrate implicit and explicit pitch memory research. Through the identification of intermediate AP performers, using tests such as the one administered in the present study, future work could for example test whether intermediate AP individuals display more accurate implicit pitch memories (e.g., differentiating in-tune and out-of-tune notes) compared to at-chance individuals, or whether implicit and explicit pitch memory abilities are independent among these populations. Such research approaches would promise to integrate explicit and implicit approaches to understanding how pitch is represented in long-term memory, as well as further clarify the distributional nature of AP.

Beyond illustrating that AP may be best described as an ability that is continuously distributed among above-chance performers, our results also challenge the view that intermediate levels of AP performance, often denigrated as "pseudo" AP, *necessarily* represents a fundamentally different process compared to "genuine" AP. In modeling the composite score, the most parsimonious representation of our data suggested by the Gaussian Mixture Model was one that only included two groups, and critically the intermediate performers were classified as belonging to the "genuine" AP distribution. Even when using a more restrictive and conservative scoring method, previously used to support a discrete view of pitch labeling ability [7,12,22], intermediate levels of pitch-labeling ability were observed in the data. Similar to the results from the composite score, these intermediate performers were much more likely to belong to the "genuine" AP distribution, not the "non-AP" distribution. In fact, the previously used threshold for AP from research groups using this operationalization arbitrarily bisected the AP distribution, highlighting the problem of using thresholds to differentiate "genuine" AP possessors from all other individuals.

In the present study, the use of uncommon timbres minimizes the possibility that the intermediate performers were only performing above chance because of extreme timbral familiarity, as can sometimes be the case for highly familiar timbres such as piano tones [56]. An additional concern with the use of complex timbres–as opposed to sine tones–in AP research is that *dynamic* spectro-temporal changes across pitch ranges (e.g., differences in the relative

power or temporal decay of harmonics) may provide an additional cue for successful note identification [57]. However, this is not applicable in the present study, as the amplitude envelope and relative power of the harmonics were fixed across all notes. Moreover, the choice to interleave octaves and provide no feedback during the assessment has been previously used to discourage the use of alternate, relative pitch strategies [18]. Furthermore, incorporating the speed of categorization into the pitch-labeling score was meant to penalize slower, deliberate response strategies that have been associated with relative pitch processing. As such, it is not likely that individuals were able to effectively use relative pitch to complete the task, at least based on the assumption that these design choices–rooted in previous research–represent valid discouragements of relative pitch strategies.

An important remaining question is thus—what factors explain performance variability among above-chance participants? One possibility is continued or specific musical expertise. Active musical training has been shown to improve pitch-production ability [58] and certain musical experiences, for example, playing a "variable *do*" instrument, can be detrimental to AP [37]. Unfortunately, given the limited nature of the questionnaire in the present study, we are unable to comment directly on how (continued) musical training relates to variability among the above-chance performers. A second possibility is (non-musical) auditory memory. For example, auditory working memory (WM) has been positively associated with the explicit learning of AP categories [34], with some high WM "pseudo" AP individuals demonstrating "genuine" AP levels of performance after eight weeks of training [35]. Furthermore, "genuine" AP possessors appear to have a larger auditory (but not visual) digit span compared to musically matched controls [59], though it is unclear how intermediate AP performers would compare to these conventional AP and non-AP groups. Third, it may be informative to use proposed biomarkers of AP-level pitch labeling ability in a more continuous framework. This approach has already displayed some promising results; for example, behavioral AP performance was found to relate significantly to the volume of white matter tracts connecting left superior temporal gyrus with left middle temporal gyrus [60].

The present results strongly argue against a fundamental *discrete dichotomy* of pitch-labeling ability; however, they do not inherently argue against a bimodal distribution of AP performance. Discrete performance implies that observations take on one of two values (e.g., "yes" or "no" with respect to possessing AP) which, given typical performance threshold for defining AP, suggests that virtually no observations should lie between these discrete groups. A more continuously distributed view of AP, in contrast, suggests that meaningful variability exists with respect to the ability to name an isolated musical note, but it also does *not* require that distribution to be uniform, unimodal, or normally distributed. While it could be argued that a bimodal distribution of AP performance still supports a dichotomy in AP, albeit loosely defined, the present data suggest that adopting such a stance would require a considerable expansion of what is typically thought to constitute AP (i.e., adopting more liberal performance thresholds that capture intermediate performance levels). Put another way, if one were to interpret the present results in a discrete framework, the data-driven approach of the present results suggest that *any* above-chance note naming performance should be included in AP investigations. In this sense, the present results fit within an emerging body of research that has demonstrated a bimodality of pitch-labeling ability, with considerable variability (i.e., a wider distribution) for individuals who perform above chance [61–63].

## 4.1 Conclusion

The present study offers important insights with respect to characterizing the distribution and proposed mechanisms of AP. Specifically, our findings suggest that "genuine" AP performers

and intermediate performers who have been characterized as having "pseudo-AP," are differentiated neither by the age in which they began musical training nor whether they speak a tonal language, the experiential factors the most commonly are thought to predict absolute pitch-labeling ability. In contrast, a Gaussian mixture modelling analysis strongly suggested that genuine-AP and pseudo-AP are merely gradations of the same ability, rather than representing discrete abilities as previously argued. Importantly, while the onset of musical training (and possibly tonal language, though this is more ambiguous in our data), do differentiate subjects with at-chance performance from subjects with above-chance pitch labeling ability, these factors do not seem to have a statistical relationship with *gradations* of absolute pitch-labeling ability.

In other words, the conceptualization of AP as a discrete ability in which subject either have near-ceiling or at-chance performance is not supported by the data, and this theoretic assumption may have misled researchers seeking to characterize the interactions between genes and experience mediating this behavioral phenotype for the better part of a century. We recommend that future studies operationalize pitch-labeling ability among above-chance performers as a continuously measured ability, rather than as a discrete trait, and the full distribution of the ability should be samples, rather than just the best and worst performers. While it may be appropriate to consider at-chance performers separately, since our results suggest that they are likely drawn from a different distribution, there appears to be meaningful variance in the gradations of pitch-labeling ability that is yet to be explained. In general, we believe that these results be taken as a cautionary tale; individual differences in perception and behavior should be considered continuously until proven otherwise, and representative samples should be used, lest progress in cognitive science be hindered by analyses framed using nonexistent dichotomies.

## Acknowledgments

The authors would like to thank Alex Huang for programming the AP assessment.

## Author Contributions

**Conceptualization:** Shannon L. M. Heald, Howard C. Nusbaum.

**Data curation:** Stephen C. Van Hedger.

**Formal analysis:** Stephen C. Van Hedger, John Veillette.

**Funding acquisition:** Howard C. Nusbaum.

**Investigation:** Stephen C. Van Hedger, Shannon L. M. Heald, Howard C. Nusbaum.

**Methodology:** Stephen C. Van Hedger, Shannon L. M. Heald.

**Project administration:** Stephen C. Van Hedger, Howard C. Nusbaum.

**Resources:** Stephen C. Van Hedger.

**Software:** John Veillette.

**Supervision:** Howard C. Nusbaum.

**Visualization:** Stephen C. Van Hedger, John Veillette.

**Writing – original draft:** Stephen C. Van Hedger, Howard C. Nusbaum.

**Writing – review & editing:** Stephen C. Van Hedger, John Veillette, Shannon L. M. Heald, Howard C. Nusbaum.

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
