## [Decision Letter · Decision Letter 0]

26 Aug 2020

PONE-D-20-13078

Revisiting discrete versus continuous models of human behavior: The case of absolute pitch

PLOS ONE

Dear Dr. Van Hedger,

Thank you for submitting your manuscript to PLOS ONE. After careful consideration, we feel that it has merit but does not fully meet PLOS ONE’s publication criteria as it currently stands. Therefore, we invite you to submit a revised version of the manuscript that addresses the points raised during the review process.

Both reviewers thought that the manuscript could be made clearer and more concise in certain sections, though those sections differed depending on reviewer (R1: Introduction; R2: Results, Conclusion, and Abstract). In general, I agree that the manuscript could be more concise and focused. For example, it takes the reader until the middle of second paragraph before we learn what the study was about (AP).

R2 also questioned one of the conclusions of the study ... that AP ability is a continuous rather than dichotomous or bimodal distribution. They point to the results of the Gaussian Mixture Modeling (Fig 5) that seem to show a bimodal distribution of AP abilities, consistent with a dichotomy.

Further suggestions that should be included in the revision:

Please include a sub-section about statistical analysis in the Methods section.Please consider using consistent terminology when referring to the opposite of a continuous distribution.Please include a histogram of AP performance (using the accuracy data shown in Fig 2A) over all listeners.

We look forward to receiving your revised manuscript.

Kind regards,

Andrew R Dykstra

Academic Editor

PLOS ONE

Journal Requirements:

Reviewers' comments:

Reviewer's Responses to Questions

**Comments to the Author**

1. Is the manuscript technically sound, and do the data support the conclusions?

Reviewer #1: Partly

Reviewer #2: Yes

2. Has the statistical analysis been performed appropriately and rigorously? 

Reviewer #1: Yes

Reviewer #2: Yes

3. Have the authors made all data underlying the findings in their manuscript fully available?

Reviewer #1: Yes

Reviewer #2: Yes

4. Is the manuscript presented in an intelligible fashion and written in standard English?

Reviewer #1: Yes

Reviewer #2: Yes

5. Review Comments to the Author

Reviewer #1: Van Hedger et al. studied pitch-labeling ability using an online assessment in a relatively large sample of participants. The study investigates to the ongoing (and long-lasting) discussion about whether absolute pitch (AP) is an all-or-nothing phenomenon or a continuously distributed ability. While the study makes an important contribution to this interesting and controversial topic, the manuscript makes certain claims that do not straightforwardly follow from the data. Furthermore, if the manuscript provided some clarifications and practical recommendations, it would appeal to a broader research community, including researchers studying the genetics or neuroscience of AP.

I would recommend a revision of the manuscript addressing the following points (and please provide line numbers to make it easier to reference parts of the manuscript):

Major points:

- Bimodal Distribution:

The study demonstrates that participants with intermediate pitch-labeling ability are more similar to participants with high levels than to participants with low levels of pitch-labeling ability. The authors take this as evidence that a “dichotomous” view of AP is not warranted (see below for a comment regarding terminology). However, the Gaussian Mixture Modeling (GMM) shows a clearly bimodal distribution suggesting a dichotomy. As the authors note in the Discussion, this dichotomy exists between chance-level performers and above chance-level performers. So, to claim that AP is a continuously distributed ability is somewhat misleading as it only seems to be continuously distributed within the group of above-chance performers. Indeed, many of the most recent AP studies used exactly this criterion (above-chance performance) to differentiate groups of AP possessors and non-possessors (e.g., Brauchli et al., 2019; Leipold et al., 2019). Thus, this framing of the results (i.e. continuous distribution) does not follow from the data.

- Introduction:

The Introduction would benefit from restructuring and significant shortening of some parts: The authors could, for example, collapse paragraphs 1 to 4 into one or two paragraphs on AP as an example for discrete vs. continuous behavior, and collapse paragraphs 5 and 6 into one paragraph explaining experiential factors influencing AP. On the contrary, a preview on what the study actually did (i.e. investigating pitch-labeling using an online assessment, mixture modeling, etc.) is completely missing. Perhaps, the authors could add a full paragraph before the Methods section explaining what they did and why they did it.

- Methods:

The authors should provide the statistical analyses already in the methods and not only in the Results section (or at least summarize it).

A clearer rationale for the group differentiation (genuine, pseudo, non) should be given, and it should be made clear that the cutoff scores are completely arbitrary (even though earlier studies also used similar, also completely arbitrary, cutoffs).

To ensure comparability with previous studies, in addition to the composite score and the conservative score, the distribution of percentage correct (or absolute number of correct trials; i.e. accuracy) should be provided. This simple and intuitive measure has been employed in many AP studies to date and is easier to interpret than the other two measures.

- Results:

The authors should provide a clear rationale for the white/black-key analyses (including Figure 3). Is this only for the validation of the composite score?

Minor points:

- Terminology: It is my impression that the authors use “discrete”, “dichotomous”, “categorical”, “all-or-nothing”, and “present or absent” synonymously, and as an antonym to “continuously distributed”. Consistent and precise terminology would help understanding the arguments that the paper tries to make.

Furthermore, I would recommend consistently using pitch-labeling ability and pitch-labeling test/task instead of AP ability/test as AP can also be assessed using other kinds of tests (production, Stroop-like, etc.). In the same vein, the mix of “pseudo” and “intermediate” is confusing.

- Methods:

When putting the sample size in context by comparing it to previous studies, the authors should also note that another study arguing for a dichotomy (Athos et al., 2007) had a much larger sample size.

Why did the authors use tones with timbre and not sine tones, given the discussion about timbral cues in pitch labeling (“absolute piano”)?

Please provide more details on the power analysis. Is this a post-hoc sensitivity analysis? Only for the GMM or other statistical analyses?

- Results

Provide the individual data points in Figures 2C and 2D instead of/or overlaying the barplots.

The statement "One thing that becomes immediately clear is that performance was highly variable and continuously distributed [...] and challenging a strict dichotomy in AP performance (cf. Athos et al., 2007)" is not appropriate at the beginning of the results section. Although I agree that the pitch-labeling performance seems highly variable and continuously distributed, we cannot determine based on visual inspection of a scatterplot if this challenges a dichotomy of AP. Distribution plots (as in Figure 5) are much more informative in this regard.

- Previous research on pitch-labeling ability

The study misses references to previous large-scale investigations of pitch-labeling abilities in the context of neuroimaging studies with over 100 participants (Brauchli et al., 2019; Leipold et al., 2019; Wengenroth et al., 2014) that show a bimodal distribution of pitch-labeling abilities, consistent with the distribution shown in the manuscript.

- Relative pitch (RP) musicians:

The manuscript does not consider the role of highly-trained musicians using RP to identify pitches. Previous research has shown that musicians without AP perform better than non-musicians in pitch-labeling tasks (e.g., Brauchli et al., 2020). Other studies even designed specific pitch-labeling tasks to prevent musicians from using RP strategies (Wengenroth et al., 2014). Some RP musicians perform better than self-reported AP possessors (e.g., Leipold et al., 2019). What role might RP play in the distribution of pitch-labeling ability?

- Conclusions:

What does “test specific hypotheses that can at least hold the hope of rejecting explicit theories” mean? Instead of speaking of “bias […] that impedes progress”, please provide clear recommendations on how future investigations (especially those on the underlying mechanisms/explanations of AP) should proceed. Group studies with more liberal criteria to include intermediate performers? How to avoid arbitrary cutoff-scores to differentiate the groups? Self-report? Correlation analyses?

References:

Athos, E. A., Levinson, B., Kistler, A., Zemansky, J., Bostrom, A., Freimer, N. B., & Gitschier, J. (2007). Dichotomy and perceptual distortions in absolute pitch ability. Proceedings of the National Academy of Sciences of the United States of America, 104(37), 14795–14800. https://doi.org/10.1073/pnas.0703868104

Brauchli, C., Leipold, S., & Jäncke, L. (2019). Univariate and multivariate analyses of functional networks in absolute pitch. NeuroImage, 189, 241–247. https://doi.org/10.1016/J.NEUROIMAGE.2019.01.021

Brauchli, C., Leipold, S., & Jäncke, L. (2020). Diminished large-scale functional brain networks in absolute pitch during the perception of naturalistic music and audiobooks. NeuroImage, 216, 116513. https://doi.org/10.1016/j.neuroimage.2019.116513

Leipold, S., Brauchli, C., Greber, M., & Jäncke, L. (2019). Absolute and relative pitch processing in the human brain: Neural and behavioral evidence. Brain Structure and Function, 224(5), 1723–1738. https://doi.org/10.1007/s00429-019-01872-2

Wengenroth, M., Blatow, M., Heinecke, A., Reinhardt, J., Stippich, C., Hofmann, E., & Schneider, P. (2014). Increased volume and function of right auditory cortex as a marker for absolute pitch. Cerebral Cortex, 24(5), 1127–1137. https://doi.org/10.1093/cercor/bhs391

Reviewer #2: General comments:

This paper provides evidence of the large spread of absolute pitch abilities which contradicts the idea of dichotomy implying that people either have it or do not have it. It also investigates relationship between age of start of musical training and proportion of tonal language speakers and find that age of onset is similar for the two above-chance groups but earlier for those groups than for the around-chance-performing group. In contrast, they did not find a significant effect of proportion of tonal language speakers. This is acknowledged in the result section but not in the abstract and conclusion. Especially in the abstract where this effect is mentioned, this seems misleading and need to be clarified. Also, the findings in terms of effect of musical training or tonal language is lacking from the conclusion.

The paper starts with a nice introduction that clearly communicates the background and motivation for the study. The paper includes a lot of statistical analysis which is nice, but the result section is far too long, and the main points are sometimes lost in all the details. I would suggest putting a lot of the details in a supplementary section to make the main story clearer. One idea is to go through the analysis using one measure (maybe put all but the most important statics in tables or supplementary materials) and then mention that the same conclusions are reached with the second measure and include the details of those results in the supplementary materials.

Finally, the conclusion is also too long and does not provide answers to all research questions of the paper. Please change that.

A general question is whether there is not an effect of number of years playing an instrument or having received music lessons? E.g., if somebody received music lessons from they were 5-6 years old, this seems likely to be less effective than if received lessons for 20 years and started when they were 5 years.

Specific comments:

P7: If the participants were not actively recruited, how did they know about the study?

P8. What do you mean with ”triangle” tones. What is the reason for including these two types of tones? Is there a reason for including more than one type? When talking about the “smooth” tones, is it the first 9 harmonics including the F0. This is not clear. Also, is this the same for the triangle tones?

Procedure: How long time did the experiment approximately take

P9, paragraph1: Did you also do the analysis without the response time? Were the conclusions the same? The MAD seems like an interesting measure in itself and there must be a lot of variation in response time that is unrelated to AP performance abilities. Some people are probably just slower than others? E.g., in the group of chance performers how much did the RT vary. For this group, shorter response time is unlikely to be related to better ability.

P11: Why did you pick the criteria of 81.3% for the gAP group?

The fact that you have continuous distribution in scores for the gAP group suggest that not everybody in the group were very near ceiling and therefore there ought to be some variance. However, the range of scores for the pAP group is larger (11-39 notes as opposed to 39-48 notes) which leaves more room for within-group variation.

P12:

I suggest replacing “lower” performance with “worse” performance

P14:

Third paragraph: When summing up the findings you ought to mention that there does not seem to be a significant effect of tonal language across groups. Or at least include this in a brief summary at the end of the previous paragraph.

It seems that you use the Baysian analysis to verify the similarity of the age and proportion of tonal language speakers, however, this purpose is not very clear. Please condense and clarify.

Why is comparison of pAP and nAP considered an alternative hypothesis? Please clarify. Don’t these results indicate that the tonal language is not widely different between groups and isn’t this similar to what you can conclude from the above analysis?

Fig1: Are these the waveform of the complex tones or for a harmonic. Are these schematics or are they actual waveforms?

Fig2: Does the “accuracy” on the x-axis of 2A refer to the proportion of correction identified? This is not clear. It seems inconsistent to here use proportions whereas percentages are used when describing the grouping in the result section.

what “proportion” does on the y-axis refer to. I presume it is the accuracy, please clarify.

Also, labels and especially legends could be improved by making them larger. It might be worth specifying what the abbreviation are in the figure text for people who just want to skim the paper.

p. 25-26: Shorten the conclusion but include conclusions to all research questions. Skip the lines before “The present study …

6. PLOS authors have the option to publish the peer review history of their article (what does this mean?). If published, this will include your full peer review and any attached files.

Reviewer #1: No

Reviewer #2: No

---

## [Author Response · Author response to Decision Letter 0]

28 Oct 2020

Please refer to the resubmission documents for a detailed response to the editor and reviewer comments that maintains formatting. The text is pasted below for posterity.

Response to Editor Comments

In addition to agreeing with the reviewers that the manuscript could be clearer and more concise, Dr. Dykstra suggested the following:

1. Please include a sub-section about statistical analysis in the Methods section.

Author’s Response: This sub-section has now been added to the Methods section.

2. Please consider using consistent terminology when referring to the opposite of a continuous distribution.

Author’s Response: We now consistently refer to the opposite of a continuous distribution as a discrete distribution. The consistent continuous-versus-discrete terminology helps to clarify the manuscript and is well grounded in prior AP research.

3. Please include a histogram of AP performance (using the accuracy data shown in Fig 2A) over all listeners.

Author’s Response: We have now added a histogram of AP performance using the accuracy data from Fig. 2A over all listeners. 

 

Response to Reviewer 1 Comments

Van Hedger et al. studied pitch-labeling ability using an online assessment in a relatively large sample of participants. The study investigates to the ongoing (and long-lasting) discussion about whether absolute pitch (AP) is an all-or-nothing phenomenon or a continuously distributed ability. While the study makes an important contribution to this interesting and controversial topic, the manuscript makes certain claims that do not straightforwardly follow from the data. Furthermore, if the manuscript provided some clarifications and practical recommendations, it would appeal to a broader research community, including researchers studying the genetics or neuroscience of AP.

I would recommend a revision of the manuscript addressing the following points (and please provide line numbers to make it easier to reference parts of the manuscript):

Major points:

- Bimodal Distribution:

1. The study demonstrates that participants with intermediate pitch-labeling ability are more similar to participants with high levels than to participants with low levels of pitch-labeling ability. The authors take this as evidence that a “dichotomous” view of AP is not warranted (see below for a comment regarding terminology). However, the Gaussian Mixture Modeling (GMM) shows a clearly bimodal distribution suggesting a dichotomy. As the authors note in the Discussion, this dichotomy exists between chance-level performers and above chance-level performers. So, to claim that AP is a continuously distributed ability is somewhat misleading as it only seems to be continuously distributed within the group of above-chance performers. Indeed, many of the most recent AP studies used exactly this criterion (above-chance performance) to differentiate groups of AP possessors and non-possessors (e.g., Brauchli et al., 2019; Leipold et al., 2019). Thus, this framing of the results (i.e. continuous distribution) does not follow from the data.

Author’s Response: We strongly agree with this interpretation, and we appreciate the feedback that this was unclear in the paper. We are arguing against the traditional dichotomy used in AP research (“genuine AP” vs. everybody else) rather than against a dichotomy in general. In the revised text, each time we state that AP is a continuously distributed ability, we have now appended a more appropriate “among above-chance performers,” except when referring to conclusions from other published papers. We also now cite the provided papers (Brauchli et al., 2019; Leipold et al., 2019) in support of the treatment of AP as more continuous – at least among individuals who demonstrate above-chance performance. We feel that this modification makes the paper significantly clearer in its conclusions, and we thank the reviewer for bringing up this point

 - Introduction:

2. The Introduction would benefit from restructuring and significant shortening of some parts: The authors could, for example, collapse paragraphs 1 to 4 into one or two paragraphs on AP as an example for discrete vs. continuous behavior, and collapse paragraphs 5 and 6 into one paragraph explaining experiential factors influencing AP. On the contrary, a preview on what the study actually did (i.e. investigating pitch-labeling using an online assessment, mixture modeling, etc.) is completely missing. Perhaps, the authors could add a full paragraph before the Methods section explaining what they did and why they did it.

Author’s Response: We have restructured the introduction as suggested. Specifically, paragraphs 1 and 4 have been merged, and paragraphs 5 and 6 have been merged. We now also include a final paragraph prior to the Method section that explicitly details the approach and rationale of the study.

- Methods:

3. The authors should provide the statistical analyses already in the methods and not only in the Results section (or at least summarize it).

This is a good suggestion, especially in light of Reviewer 2’s comments that the results section was getting too long. We have moved the technical descriptions and brief justifications for the analyses into the methods section, in which they now have their own subsection (2.5). 

4. A clearer rationale for the group differentiation (genuine, pseudo, non) should be given, and it should be made clear that the cutoff scores are completely arbitrary (even though earlier studies also used similar, also completely arbitrary, cutoffs).

Author’s Response: We now explicitly note that the groups are arbitrary and provide a clearer explanation in the first paragraphs of Section 2.5, which is quoted directly below for convenience. Since the difference between pseudo-AP and non-AP is determined by a chance-level threshold, the only truly arbitrary choice is the cutoff between genuine- and pseudo-AP. Of course, our whole argument is eventually that splitting those above-chance subjects into two groups is not justified by the data, which we illustrate using mixture modelling, but we agree it was a good idea to make this clear earlier. 

Our first analysis goal was to characterize our data using the (often arbitrary) performance thresholds that are characteristic of AP research so that we could subsequently compare these groupings to a more data-driven categorization. To this end, we broke subjects into three (arbitrary) groups based on their pitch classification accuracies. The break between chance-level, non-AP performers (nAP) and above-chance performers was set at 11 of 48 correctly-identified notes (22.9%) because achieving this level of accuracy or higher significantly differed from the chance estimate of 4 of 48 notes (Binomial Test: p = .0003), even when using a false discovery rate alpha correction (FDR q = .0007). 

The above-chance performers were further subdivided into pseudo-AP (pAP) performers, whom the AP literature would traditionally not consider performing strongly enough to have AP, and genuine-AP (gAP) performers. The threshold between these two groups was set at 39 of 48 correctly identified notes (81.3%) – a value that was based on prior research. For example, Deutsch et al. (2006) used a similar threshold (85%) and operationalized performance both liberally (including semitone errors) and conservatively (only including correct classifications). Furthermore, Miyazaki (1989) only tested self-identified AP possessors; however, in this study, the mean accuracy for complex tones (like those used in the present study) was 80.4%. As such, while the placement of any threshold is likely to be arbitrary (assuming a continuous distribution of performance), our selected threshold is grounded in prior research. Whether it is appropriate to separate these two groups at all is tested later in our analyses.

5. To ensure comparability with previous studies, in addition to the composite score and the conservative score, the distribution of percentage correct (or absolute number of correct trials; i.e. accuracy) should be provided. This simple and intuitive measure has been employed in many AP studies to date and is easier to interpret than the other two measures.

Author’s Response: This is also a good idea. A histogram of accuracy is now shown in Figure 2B. We do not include accuracy in the Gaussian Mixture modelling because accuracy is Beta distributed (since it is [0, 1] bounded) rather than Gaussian distributed, and the Beta-Binomial distribution does not necessarily have a closed-form maximum likelihood parameter estimate, which complicates the Maximization step of the Expectation-Maximization fitting procedure for mixture models substantially.

- Results:

6. The authors should provide a clear rationale for the white/black-key analyses (including Figure 3). Is this only for the validation of the composite score?

Author’s Response: It is mostly to validate the sensitivity of the test, but it does have the added benefit of showing that the pseudo-AP subjects show the same bias considered characteristic of genuine-AP subjects. The justification regarding sensitivity is now stated in Methods 2.5 and again when results are stated. The added benefit is pointed out in the caption of Figure 3.

Minor points

- Terminology:

7. It is my impression that the authors use “discrete”, “dichotomous”, “categorical”, “all-or-nothing”, and “present or absent” synonymously, and as an antonym to “continuously distributed”. Consistent and precise terminology would help understanding the arguments that the paper tries to make.

Author’s Response: We have replaced nearly every instance of “dichotomous” with “discrete.” In some cases, we do leave less-used phrases like “categorical” or “all-or-nothing” in the text where we believe it clarifies our argument. Specifically, we think it is important to clarify in some cases, especially near the beginning of the manuscript, what we mean by “discrete,” though we take the point that consistent terminology is critical and have made appropriate changes to that effect. 

8. Furthermore, I would recommend consistently using pitch-labeling ability and pitch-labeling test/task instead of AP ability/test as AP can also be assessed using other kinds of tests (production, Stroop-like, etc.). In the same vein, the mix of “pseudo” and “intermediate” is confusing.

Author’s Response: We agree with this suggestion. We have made this change in almost all instances. The notable exception is that, when we refer to “genuine-AP ability,” we would like to keep the words “genuine” and “AP” next to each other to make it easier for the reader to connect that phrase to the genuine-AP group. This only occurs a few times in the text. There is also one instance in which we refer to “different operationalizations of AP ability,” but we think this is still appropriate by the logic of the Reviewer’s suggestion since it acknowledges that there are multiple operationalizations. 

- Methods:

9. When putting the sample size in context by comparing it to previous studies, the authors should also note that another study arguing for a dichotomy (Athos et al., 2007) had a much larger sample size.

Author’s Response: Good point. This has been noted on Line 302.

10. Why did the authors use tones with timbre and not sine tones, given the discussion about timbral cues in pitch labeling (“absolute piano”)?

Author’s Response: This is a good question. We did not deliberately select complex tones over sine tones for any strong reasons. However, we do not think the use of (unfamiliar) complex tones in the present research is problematic. This is because our understanding of “absolute piano” is that listeners can use extreme timbral familiarity – in conjunction with the dynamic changes in harmonics across the piano range – to make accurate pitch-label judgments. For example, when detailing why there might be meaningful differences in AP accuracy as a function of timbre, Takeuchi and Hulse (1993) suggest two possibilities. First, some timbres might be much more familiar than others (e.g., piano versus computer-synthesized tones). Second, “variations in timbre over changes in pitch may provide cues to pitch” (p. 351). In other words, the relative power of harmonic / inharmonic frequencies, as well as amplitude envelope, are often dynamic across pitches for real instruments like the piano. The worry, then, is that these subtle timbral changes across pitches can be used as an additional cue to determine pitch identity. Indeed, this is precisely how Lockhead and Byrd (1981) frame their influential comparison of sine and piano tones. In their Introduction, Lockhead and Byrd write “the harmonic development of piano notes is not constant throughout the range. There are marked timbre differences between notes in the bottom two octaves and the remaining notes; also, there are very different decay characteristics between piano notes in the top two octaves and the remainder of the range. If these or other such factors can be used in identifying a note, then the piano is an undesirable source of test tones” (p.387). These problems with highly familiar and dynamic tones, like piano tones, can lead to “absolute piano” (as the reviewer suggests). Thus, the use of unfamiliar computer-generated tones - with static and consistent harmonic spectra across all test notes – satisfies the potential concerns of using complex tones.

Finally, from a more pragmatic perspective, we modeled many aspects of our study approach after Bermudez and Zatorre (2009), who also reported a distribution of pitch-labeling performance and used complex (non-instrumental) tones. 

11. Please provide more details on the power analysis. Is this a post-hoc sensitivity analysis? Only for the GMM or other statistical analyses?

Author’s Response: We appreciate this comment as it made us realize the power description was unclear. The power analysis was a priori and was based on the null hypothesis significance testing reported in the first portion of the results (e.g., the power to detect a difference in the age of music onset when comparing the three groups). We have now revised this description to read:

Moreover, based on an a priori power analysis using (G*Power), the present sample size is sufficiently powered (β = .8) to detect medium effect sizes even when dividing the sample into three groups; i.e., genuine, intermediate, and non-AP (f = 0.254).. The planned analysis that was used in the power analysis was a one-way ANOVA (e.g., assessing differences for age of music onset across groups); the power analysis did not specifically inform the Gaussian Mixture Modeling described in Section 3.4.

- Results

12. Provide the individual data points in Figures 2C and 2D instead of/or overlaying the barplots.

Author’s Response: We have now included all individual points in Figure 2C, which has now been moved to become Figure 2B. 

We did not do the same for Figure 2D, because the tonal language data is binary (one if the subject speaks a tonal language, zero otherwise). Therefore, the summary statistic (proportion) provides as complete a picture of the data as would plotting the individual points – adding the points, we think, would be visually distracting without adding more information. However, we do take the point that bar plots can be perceptually misleading, so we have replaced the bars with a minimalistic standard error plot. 

The individual data points for both age of music onset and tone language experience are additionally now included in Figure 5, plotted against both the composite score and the conservative measure. We could redo Figures 2B and 2D to resemble these new Figures in 5 more closely, if the editors and reviewers find that preferable. 

13. The statement "One thing that becomes immediately clear is that performance was highly variable and continuously distributed [...] and challenging a strict dichotomy in AP performance (cf. Athos et al., 2007)" is not appropriate at the beginning of the results section. Although I agree that the pitch-labeling performance seems highly variable and continuously distributed, we cannot determine based on visual inspection of a scatterplot if this challenges a dichotomy of AP. Distribution plots (as in Figure 5) are much more informative in this regard.

Author’s Response: We have removed this statement. 

- Previous research on pitch-labeling ability

14. The study misses references to previous large-scale investigations of pitch-labeling abilities in the context of neuroimaging studies with over 100 participants (Brauchli et al., 2019; Leipold et al., 2019; Wengenroth et al., 2014) that show a bimodal distribution of pitch-labeling abilities, consistent with the distribution shown in the manuscript.

Author’s Response: We appreciate these references; they are now incorporated into the manuscript to support the distributional findings of the present study. Specifically, in the Discussion, we now write:

In this sense, the present results fit within an emerging body of research that has demonstrated a bimodality of pitch-labeling ability, with considerable variability (i.e., a wider distribution) for individuals who perform above chance (Brauchli, Leipold, & Lutz, 2019; Leipold, Greber, Sele, & Lutz, 2019; Wengenroth et al., 2014).

- Relative pitch (RP) musicians:

15. The manuscript does not consider the role of highly-trained musicians using RP to identify pitches. Previous research has shown that musicians without AP perform better than non-musicians in pitch-labeling tasks (e.g., Brauchli et al., 2020). Other studies even designed specific pitch-labeling tasks to prevent musicians from using RP strategies (Wengenroth et al., 2014). Some RP musicians perform better than self-reported AP possessors (e.g., Leipold et al., 2019). What role might RP play in the distribution of pitch-labeling ability?

Author’s Response: This is an interesting comment, and we appreciate the reviewer bringing up this issue and providing recent citations (which are now incorporated in the manuscript). In the Discussion, we have revised our discussion of the design choices we use to discourage relative pitch strategies (printed below):

The choice to interleave octaves and provide no feedback during the assessment has been previously used to discourage the use of alternate, relative pitch strategies (e.g., see Bermudez & Zatorre, 2009). Furthermore, incorporating the speed of categorization into the pitch-labeling score was meant to penalize slower, deliberate response strategies that have been associated with relative pitch processing. As such, it is not likely that individuals were able to effectively use relative pitch to complete the task, at least based on the assumption that these design choices – rooted in previous research – represent valid discouragements of relative pitch strategies. 

We really appreciate this comment, as it has made us think about the role of RP in pitch-labeling ability more critically. In short, while we designed the present research study in a manner that was meant to discourage effective RP strategies, it is entirely possible that highly trained musicians, with a stable reference note (or notes), were able to use some combination of AP and RP strategies, provided the RP judgments were sufficiently quick and the AP references were sufficiently stable. 

- Conclusions:

16. What does “test specific hypotheses that can at least hold the hope of rejecting explicit theories” mean? Instead of speaking of “bias […] that impedes progress”, please provide clear recommendations on how future investigations (especially those on the underlying mechanisms/explanations of AP) should proceed. Group studies with more liberal criteria to include intermediate performers? How to avoid arbitrary cutoff-scores to differentiate the groups? Self-report? Correlation analyses?

Author’s Response: We now include specific recommendations instead of the previous (admittedly vague) statement. The relevant text is below.

We recommend that future studies operationalize pitch-labeling ability as a continuously measured ability, rather than as a discrete trait, and the full distribution of the ability should be samples, rather than just the best and worst performers. While it may be appropriate to consider at-chance performers separately, since our results suggest that they are likely drawn from a different distribution, there appears to be meaningful variance in the gradations of pitch-labeling ability that is yet to be explained.

Response to Reviewer 2 Comments

1. This paper provides evidence of the large spread of absolute pitch abilities which contradicts the idea of dichotomy implying that people either have it or do not have it. It also investigates relationship between age of start of musical training and proportion of tonal language speakers and find that age of onset is similar for the two above-chance groups but earlier for those groups than for the around-chance-performing group. In contrast, they did not find a significant effect of proportion of tonal language speakers. This is acknowledged in the result section but not in the abstract and conclusion. Especially in the abstract where this effect is mentioned, this seems misleading and need to be clarified. Also, the findings in terms of effect of musical training or tonal language is lacking from the conclusion.

Author’s Response: We have edited the abstract and rewritten the conclusion. As we discuss more below, we did find a difference between the highest and lowest performers in tonal language use, so the original statement in the abstract is technically correct, but frequentist and Bayesian analysis disagreed on that front, so we agree we should state that less conclusively in the abstract. The abstract text about tonal language and music experience is now as below.

Consistent with prior research, individuals who performed at-chance (non-AP) reported beginning musical instruction much later to the near-perfect AP participants, and the highest performers were more likely to speak a tonal language than were the lowest performers (though this effect was not as statistically robust as one would expect from prior research).

2. The paper starts with a nice introduction that clearly communicates the background and motivation for the study. The paper includes a lot of statistical analysis which is nice, but the result section is far too long, and the main points are sometimes lost in all the details. I would suggest putting a lot of the details in a supplementary section to make the main story clearer. One idea is to go through the analysis using one measure (maybe put all but the most important statics in tables or supplementary materials) and then mention that the same conclusions are reached with the second measure and include the details of those results in the supplementary materials.

Author’s Response: We have shortened the results section by moving the technical details of our analyses to the methods. We hope that this ameliorates the problem. We are also happy to move some analyses (such as the white-key advantage analysis) to a supplementary information section if the referees feel that the results section is still too lengthy. However, we have kept them in for now since we feel that is important to compare results across operationalizations of AP directly to build our argument that AP should be considered as a graded ability regardless of how you measure it. We do appreciate the suggestions on how to shorten things further though, and we are happy to oblige if the editor agrees.

3. Finally, the conclusion is also too long and does not provide answers to all research questions of the paper. Please change that.

Author’s Response: We have rewritten the conclusion based on reviewer recommendations.

4. A general question is whether there is not an effect of number of years playing an instrument or having received music lessons? E.g., if somebody received music lessons from they were 5-6 years old, this seems likely to be less effective than if received lessons for 20 years and started when they were 5 years.

Author’s Response This is a good question, and unfortunately, we do not have the data in the present study to answer this directly (as the simple questionnaire did not ask about individuals’ specific musical training experiences beyond the age of beginning instruction). Generally speaking, the emphasis in AP research is on the age at which musical instruction commences; however, active musical training has also been associated with aspects of AP performance (e.g., Dohn et al., 2014 https://doi.org/10.1525/mp.2014.31.4.359). As such, we think this is an excellent question that is unfortunately beyond the scope of the present work. 

We now explicitly mention this limitation in the Discussion. Specifically, we write:

An important remaining question is thus - what factors explain performance variability among above-chance participants? One possibility is continued or specific musical expertise. Active musical training has been shown to improve AP pitch-production ability (Dohn, Garza-Villarreal, Ribe, & Wallentin, 2014) and certain musical experiences, for example, playing a “variable do” instrument, can be detrimental to AP (Wilson et al., 2012). Unfortunately, given the limited nature of the questionnaire in the present study, we are unable to comment directly on how (continued) musical training relates to variability among the above-chance performers.

Specific comments:

5. P7: If the participants were not actively recruited, how did they know about the study?

The Wall Street Journal wrote about our test in an article (this was a big stroke of luck for us), resulting in a big surge of participation by people wanting to know if they had AP. This is stated in the manuscript, which we have reproduced below for convenience. We also noted it a second time on Lines 296-297. 

Approximately half of the 195 participants (48.6%) completed the study within a one-week period (from June 11, 2017 to June 16, 2017). We attribute this spike in participation to a Wall Street Journal article published on June 11, 2017 (Mitchell, 2017) that provided a link to the online study. 

6. P8. What do you mean with “triangle” tones. What is the reason for including these two types of tones? Is there a reason for including more than one type? When talking about the “smooth” tones, is it the first 9 harmonics including the F0. This is not clear. Also, is this the same for the triangle tones?

Author’s Response: The waveform of a triangle tone is shown in Figure 1. These, along with smooth tones, are sometimes used in AP research as a middle ground between sine tones, which in principle have no harmonics, and natural notes from instruments that subjects may have some implicit memory of from some other musical context. In contrast, the “smooth” and “triangle” tones have some energy in the harmonics but nonetheless an unfamiliar timbre, so subjects can leverage harmonic information, which is a valid pitch cue, as they would in a natural setting but should not be able to leverage their experience with a particular instrument to arrive at an answer through a relative-pitch type strategy. Further rationale for including these kinds of tones (e.g., as opposed to piano tones) is provided in response to comment #10 from Reviewer 1.

With respect to the harmonics in the smooth tone, we appreciate the reviewer pointing out the ambiguity in our description. The smooth tones included 9 harmonics in addition to the fundamental frequency (i.e., each tone had 10 frequency components). This is now clarified in the manuscript:

The smooth tones were generated using the “inverted sine” option in Adobe Audition (Adobe Systems: San Jose, CA), which did not result in a true sinusoid but rather a complex tone with 9 harmonics (in addition to the fundamental frequency) and an approximate 11dB reduction for each harmonic.

7. Procedure: How long time did the experiment approximately take

Author’s Response: This is a good question. Based on the timestamps, most participants completed the AP assessment quite quickly (5.5 minutes on average). We have now included this information at the end of the Procedure (reprinted below).

The AP assessment took participants approximately five and a half minutes to complete, and most participants completed the entire procedure (from consent to the feedback screen) in under 10 minutes.

8. P9, paragraph1: Did you also do the analysis without the response time? Were the conclusions the same? The MAD seems like an interesting measure in itself and there must be a lot of variation in response time that is unrelated to AP performance abilities. Some people are probably just slower than others? E.g., in the group of chance performers how much did the RT vary. For this group, shorter response time is unlikely to be related to better ability.

Author’s Response: We did do the analyses with the MAD alone, and the conclusions were the same. They are not included in the manuscript, since the MAD did not seem normally distributed since it is bound in the interval [0, 6]. Results from other operationalizations held if we ignored this assumption violation and blindly applied Gaussian Mixture Modelling, but fitting a mixture of gamma distributions still seemed like it would have been more appropriate. This seemed needlessly complicated to get into, since MAD results did not seem to be meaningfully different and most readers would not be as familiar with the gamma distribution. 

9. P11: Why did you pick the criteria of 81.3% for the gAP group?

The fact that you have continuous distribution in scores for the gAP group suggest that not everybody in the group were very near ceiling and therefore there ought to be some variance. However, the range of scores for the pAP group is larger (11-39 notes as opposed to 39-48 notes) which leaves more room for within-group variation.

Author’s Response: We ended up selecting 81.3% (39 of 48 correctly identified notes) based on prior thresholds used in AP research. (For a more detailed response, refer to our response to comment #4 from Reviewer 1.) To briefly summarize, this threshold was in line with prior AP investigations (e.g., the 85% threshold used by Deutsch and colleagues, though the Deutsch assessment used piano tones, which could possibly inflate performance due to timbral familiarity). Additionally, Miyazaki tested self-identified AP possessor performance across sine tone, complex tones (like those in the present study), and piano tones, finding mean accuracy levels of 80.4% for the complex tones. However, despite our desire to ground this threshold in prior literature, we completely agree that any threshold – particularly in the context of separating the gAP and pAP groups – is arbitrary. We now describe the thresholds justification (as well as the inherent arbitrariness in threshold determination) in Section 2.5 (quoted below):

For example, Deutsch et al. (2006) used a similar threshold (85%) and operationalized performance both liberally (including semitone errors) and conservatively (only including correct classifications). Furthermore, Miyazaki (1989) only tested self-identified AP possessors; however, in this study, the mean accuracy for complex tones (like those used in the present study) was 80.4%. As such, while the placement of any threshold is likely to be arbitrary, our selected threshold is grounded in prior research. Whether it is appropriate to separate these two groups at all is tested later in our analyses.

P12:

10. I suggest replacing “lower” performance with “worse” performance

Author’s Response: We have made this change. “Lower” is still used when referring to particular scores (e.g. “they had a lower composite score”) but it is usually clarified immediately after whether lower corresponds to better or worse performance, since we understand this can get confusing as low scores on the composite measure correspond to good performance but one the other measures correspond to bad performance. 

P14:

11. Third paragraph: When summing up the findings you ought to mention that there does not seem to be a significant effect of tonal language across groups. Or at least include this in a brief summary at the end of the previous paragraph.

Author’s Response: We touch on this more in the discussion of our Bayes Factor analysis below, but briefly – there was in fact a significant difference in tone language experience between non-AP and genuine-AP groups, replicating previous results. (Even though the full model for tone language was not significant, which we think is what you are referring to.) Importantly, the Bayesian analysis did not support there being a difference between nAP and gAP groups, contradicting the frequentist analysis. Since the Bayesian and frequentist analyses disagreed, we did not think tone language results were sufficiently conclusive to dwell on them in the parts of the paper readers might skim. Rather, we reported both analyses in the results section so that interested readers could draw their own conclusions. 

12. It seems that you use the Bayesian analysis to verify the similarity of the age and proportion of tonal language speakers, however, this purpose is not very clear. Please condense and clarify.

Author’s Response: This is indeed why we use the Bayesian analysis. We now justify this more thoroughly in two places – once in Methods 2.5, the other in Results where the Bayes Factors are reported. The latter instance is quoted below for convenience.

These results suggest that the pAP group and the gAP groups had comparable ages of beginning musical instruction and proportions of tone language proportions, although these findings must be interpreted cautiously as they rest on the acceptance of a null hypothesis. To facilitate appropriate interpretation, since a null result here could be theoretically important but the regression above analyses provides us with little information about the posterior probability the null hypothesis is correct, we computed Bayes Factors (BF01) to assess evidence in favor of the null.

13. Why is comparison of pAP and nAP considered an alternative hypothesis? Please clarify. Don’t these results indicate that the tonal language is not widely different between groups and isn’t this similar to what you can conclude from the above analysis?

Author’s Response: We are not entirely sure what is meant here. The null hypothesis, statistically speaking, is that there is no difference between groups, and the alternative hypothesis in a two-tailed test is that there is such a difference. But the assessment that tonal language is not widely different between groups is indeed a sound interpretation, since the data were 4.03 times more likely to occur under the null hypothesis (no difference between groups) than under the alternative hypothesis (a true difference between groups). 

This is similar to what was concluded in the preceding frequentist analysis, except the Fisher’s exact test found that there indeed was a significant difference in tone language experience between the non-AP and genuine-AP groups. This can happen sometimes for large samples when the p-value is “significant” but barely below the threshold, since the probability of seeing a p-value near 0.05 (as opposed to much lower) is actually very low if n is large – sometimes so low that it is actually more likely to see such a p-value under the null hypothesis! (See the link below for an excellent blog post about this statistical curiosity.) A Bayesian analysis cares about the relative probability of seeing the data under the null and alternative hypotheses, so this is one case in which frequentist and Bayesian analyses disagree. This, of course, touches on a broader issue surrounding the interpretation of p-values and a centuries old debate about whether we should use frequentist or Bayesian statistics. Our stance is that we only like to confidently state results when the two schools agree, and otherwise we will report both analyses in the results but not state anything definitive in the parts of the paper that are more easily skimmed, which is why we had not initially dwelled on the tone language data in the abstract and conclusion. 

https://daniellakens.blogspot.com/2015/03/how-p-value-between-004-005-equals-p.html

14. Fig1: Are these the waveform of the complex tones or for a harmonic. Are these schematics or are they actual waveforms?

Author’s Response: These are the actual waveforms for two exemplary tones (although “zoomed in” as shown by the time scale on the x-axis). Computing the Fourier transform of the waveforms shown on the left produce the power spectra shown on the right, where the fundamental frequency and the harmonics are visible as peaks of the spectrum. We have now added in the figure caption that these plots were generated from actual tones used in the assessment (A3; 220Hz).

15. Fig2: Does the “accuracy” on the x-axis of 2A refer to the proportion of correction identified? This is not clear. It seems inconsistent to here use proportions whereas percentages are used when describing the grouping in the result section.

Author’s Response: Yes, that is what we meant. We have changed the axis label to reflect the percentage of correct answers rather than the proportion for consistency, and we specify in the figure text that this refers to the percentage of trials answered correctly. 

16. what “proportion” does on the y-axis refer to. I presume it is the accuracy, please clarify.

Author’s Response: This is now clarified in the figure text. 

17. Also, labels and especially legends could be improved by making them larger. It might be worth specifying what the abbreviation are in the figure text for people who just want to skim the paper.

Author’s Response: We have redone the figures with larger text. Full group names, rather than just abbreviations, are now stated in the figure text. 

18. p. 25-26: Shorten the conclusion but include conclusions to all research questions. Skip the lines before “The present study …

Author’s Response: The conclusion has been shortened appropriately. We then added a paragraph with specific recommendations as suggested by Reviewer 1, which made it somewhat longer again, but the length is now justified by useful content rather than by needlessly restated information.

---

## [Decision Letter · Decision Letter 1]

8 Dec 2020

Revisiting discrete versus continuous models of human behavior: The case of absolute pitch

PONE-D-20-13078R1

Dear Dr. Van Hedger,

We’re pleased to inform you that your manuscript has been judged scientifically suitable for publication and will be formally accepted for publication once it meets all outstanding technical requirements.

Kind regards,

Lutz Jäncke, PhD

Academic Editor

PLOS ONE

Additional Editor Comments (optional):

The authors have done a very good job in adapting the manuscript. One original reviewer already accepted the manuscript. While looking at the added new references from the Jancke group (which has published quite a lot on absolute pitch) the authors should check their citations (the two are actually wrong). I have listed the correct references below.

Brauchli, C., Leipold, S., & Jäncke, L. (2019) Univariate and multivariate analyses of functional networks in absolute pitch. Neuroimage, 189, 241–247.

Leipold, S., Greber, M., Sele, S., & Jäncke, L. (2019) Neural patterns reveal single-trial information on absolute pitch and relative pitch perception. Neuroimage, 200, 132–141.

Reviewers' comments:

Reviewer's Responses to Questions

**Comments to the Author**

1. If the authors have adequately addressed your comments raised in a previous round of review and you feel that this manuscript is now acceptable for publication, you may indicate that here to bypass the “Comments to the Author” section, enter your conflict of interest statement in the “Confidential to Editor” section, and submit your "Accept" recommendation.

Reviewer #1: All comments have been addressed

2. Is the manuscript technically sound, and do the data support the conclusions?

Reviewer #1: Yes

3. Has the statistical analysis been performed appropriately and rigorously? 

Reviewer #1: Yes

4. Have the authors made all data underlying the findings in their manuscript fully available?

Reviewer #1: Yes

5. Is the manuscript presented in an intelligible fashion and written in standard English?

Reviewer #1: Yes

6. Review Comments to the Author

Reviewer #1: - The authors did a great job revising their manuscript. I especially liked how they clarified their interpretation of the data. All of my previous points were properly addressed, and responses were given in great detail, thank you. This paper represents an important contribution to the field of absolute pitch-related research. I recommend publication of the manuscript. Please correct the details below though.

- Line 409: Figure[s] 5A is referenced, however, this figure shows the histogram using the composite score

- Line 469: The reference to the figure should read Figure 5D. Relatedly, Figure 5B, C, E, and F are not referenced or discussed in the main text of the manuscript (maybe it is an old version of the figure?).

- Line 632: The last author’s name in Brauchli et al. (Line 732 in the references) and Leipold et al. (Line 783 in the references) is Jäncke, and not Lutz.

- Line 861: Similarly, the last author of Wengenroth et al., Peter Schneider is not included in the reference.

7. PLOS authors have the option to publish the peer review history of their article (what does this mean?). If published, this will include your full peer review and any attached files.

Reviewer #1: No

---

## [Editor Report · Acceptance letter]

10 Dec 2020

PONE-D-20-13078R1 

Revisiting discrete versus continuous models of human behavior: The case of absolute pitchRevisiting discrete versus continuous models of human behavior: The case of absolute pitch 

Dear Dr. Van Hedger:

I'm pleased to inform you that your manuscript has been deemed suitable for publication in PLOS ONE. Congratulations! Your manuscript is now with our production department. 

Kind regards, 

on behalf of

Prof Lutz Jäncke 

Academic Editor

PLOS ONE